# PortUrb: A Performance Portable, High-Order, Moist Atmospheric Large Eddy Simulation Model with Variable-Friction Immersed Boundaries

Matthew Norman[1], Muralikrishnan Gopalakrishnan Meena[1], Kalyan Gottiparthi[1],
Nicholson Koukpaizan[1], and Stephen Nichols[1]

[1]Oak Ridge National Laboratory, 1 Bethel Valley Road, Oak Ridge, TN 37830

**Correspondence:** Matthew Norman (normanmr@ornl.gov)

**Abstract.** This paper introduces "portUrb": a moist, compressible, non-hydrostatic atmospheric Large Eddy Simulation model that aims for portability, performance, accuracy, simplicity, readability, robustness, extensibility, and ensemble capabilities. Additionally, there is an emphasis on free-slip immersed boundaries with surface friction to account for urban building geometries. Coded in portable C++ with high-order Weighted Essentially Non-Oscillatory (WENO) numerics, this study investigates the behavior of portUrb under atmospheric boundary layer, supercell, and urban scenarios. PortUrb matches experimental observations and model comparisons closely under several test cases in mean and turbulent statistics. It also provides physically realizable flow through complex building geometries from a portion of Manhattan without needing to pre-process or smooth the building geometry.

## 1 Introduction

Large Eddy Simulation (LES) has been an effective tool in simulating turbulent flows in complex scenarios for decades (Mason, 1994; Lesieur and Metais, 1996; Stoll et al., 2020). This study focuses on what would, atmospherically speaking, often be called "microscale" flow (Liu et al., 2012; Zhang et al., 2016), resolving grid spacings in the range $[1, 100]$ meters. This resolution is important for understanding very fine-scaled boundary layer flows with applications to wind energy (Mehta et al., 2014), urban flow (Letzel et al., 2008; Xie and Castro, 2009), and vegetation canopy flows (Shaw and Schumann, 1992; Dupont and Brunet, 2008) among other potential applications. It is a challenging scale to simulate due to the coupled effects of large-scale forcing

such as incoming weather patterns as well as extremely small-scale forcing such as plant morphology and surface and building material roughness with complex geometries.

There are a number of effective models in this type of regime in literature including the Weather Research and Forecasting (WRF) model's LES configuration (Skamarock et al., 2019), the Parallelized Large-Eddy Simulation Model (PALM) (Maronga et al., 2015), the Dutch Atmospheric Large-Eddy Simulation (DALES) (Heus et al., 2010), the System for Atmospheric Modeling (SAM) (Khairoutdinov and Kogan, 2000; Lyngaas et al., 2022), the Cloud Model 1 (CM1) (Bryan and Fritsch, 2002), the Regional Atmospheric Modeling System (RAMS) (Cotton et al., 2003), The UCLA LES model (Stevens et al., 1998), the Advanced Regional Prediction System (ARPS) (Droegemeier, 2003), the Eulerian-Lagrangian (EULAG) model (Prusa et al., 2008), the MicroHH model (Van Heerwaarden et al., 2017), the PyCLES model (Pressel et al., 2015), the Simulator fOr Wind Farm Applications (SOWFA) (Fleming et al., 2014), the FastEddy model (Sauer and Muñoz-Esparza, 2020; Muñoz-Esparza et al., 2020, 2022), OpenFOAM (Jasak et al., 2007), the UK Met Office NERC Cloud (MONC) model (Brown et al., 2015), and the Energy Research and Forecasting (ERF) model (Almgren et al., 2023).

Existing models cover a wide range of capabilities and applications including wildfires, urban flow, canopy flow, complex terrain flow, shallow and deep convection, complex boundary layers, windfarms, and pollutant dispersion. They cover a lot of different formulations including Eulerian and Lagrangian formulations, different thermodynamic formulations, moisture inclusion approaches, acoustic representations, immersed boundary approaches, numerical discretizations, gridding and variable staggering practices, and turbulent mixing approaches. There are also many different coding practices represented. Languages include Fortran, C++, and Python. Some are CPU-threaded with directives (Bronevetsky and De Supinski, 2007). Some are accelerated for use on Graphics Processing Units (GPUs) with CUDA (Cook, 2012), directives (Chandrasekaran and Juckeland, 2017; Bronevetsky and De Supinski, 2007), or C++ portability layers (Trott et al., 2021; Norman et al., 2023a; Zhang et al., 2019). Each comes with custom extensions such as advanced microphysics or radiation schemes and various parameterizations for added applications (wind turbines, urban roughness, fire, aerosol chemistry, etc.).

The goal of this study is to introduce and investigate the properties of a very high-order-accurate moist atmospheric LES model. The model was implemented with an emphasis on simplicity, readability, portability, and extensibility. The desire is to create a model that: (1) runs portably and efficiently (in all parts of the code) on CPUs and accelerators such as GPUs in a manner extensible to future hardware by using the Kokkos C++ portability library; (2) emphasizes readability in all portions of the code for clarity and reliability (clear looping, clear array indexing, clear operations, etc.); (3) has a simple and unambiguous coupler state that is easily augmented with new modules; (4) has embedded boundaries that are simple, resilient, and able to impose roughness on tangential flow; (5) runs stably for any configuration; and (6) has an easy to use ensemble capability to enable running many different configurations simultaneously in a single executable. With these features in mind, the ultimate goal of this code is to create a rapid prototyping environment for creating surrogate models of different microscale atmospheric processes. The simplicity, portable GPU acceleration, and ensemble capabilities enable rapid prototyping. The additive representation of complex physics through modular processes (all tied to the coupler state) enables enough realism to create physically meaningful surrogate models.

The pathway to producing the model investigated in this study, called "portUrb" (Portable Urban model), is mainly two-fold. First, a series of numerical discretization studies (Norman et al., 2011; Norman, 2014; Norman and Larkin, 2020; Norman, 2021; Norman et al., 2023b) has led to the current choice of discretization: a collocated, upwind (Riemann solver-based), high-order-accurate Finite-Volume method using Weighted Essentially Non-Oscillatory (WENO) limiting (Liu et al., 1994; Feng et al., 2012) and a Strong Stability Preserving Runge–Kutta (SSPRK) discretization in time (Gottlieb et al., 2009). Also, progressive additions to a mini-application called "miniWeather" (Norman, 2020)[1] intended for hands-on education in High Performance Computing (HPC) programming has eventually led to portUrb, which provides realistic simulations while retaining much of the simplicity.

We wish to summarize the motivations for numerical discretizations chosen in this study. In doing so, we want to emphasize that discretizations form a very complex landscape with many tradeoffs, and we do not necessarily cast this study's choices as superior in any objective way. The use of a collocated grid simplifies the grid indexing and code layout, but it also introduces instabilities that would otherwise be ameliorated by different grid staggering choices. Therefore, upwind fluxes are used at cell edges in order to introduce dissipation that naturally scales with the accuracy of the reconstructions (Norman et al., 2023b). It was found in Norman et al. (2023b) that this can be performed in an inexpensive manner by separating acoustic and advective concerns in the upwind calculations. WENO limiting, while certainly expensive, has an advantage on GPUs in the sense that the extra computations required are largely performed on-chip with memory that has already been fetched from Dynamic Random Access Memory (DRAM), reducing the relative cost on GPUs. It also reduces oscillations near immersed boundaries and sharp hydrometeor fronts in moist convective simulations.

While fully-discrete Arbitrary DERivatives in space and time (ADER) time discretizations using Differential Transforms (DTs) have been investigated in the past (Norman and Finkel, 2012; Norman and Larkin, 2020; Norman, 2013a), this particular approach to ADER discretization is most economically applied in a fully dimensionally split manner. It has been found through experimentation that less dissipative results are obtained by using multi-stage integrators with dimensional splitting inside each stage. Therefore, a Runge-Kutta method is used here: specifically a SSPRK method that maintains the non-oscillatory properties of the underlying spatial operator. Finally, high-order reconstruction is desired for the same reason that WENO limiting is desired. On GPUs, the extra computational cost is ameliorated by the significant compute throughput of GPU devices, and most of the computations are performed on data that is already on-chip. While it increases overall runtime, it does so significantly less than the factor by which computations are increased.

The present paper focuses on evaluating model behavior in test cases. The mathematical formulation and numerical discretizations are provided in section 2; numerical experiments are described, performed, and discussed in section 3; and conclusions are drawn with a discussion of future work in section 4.

---

[1] https://github.com/mrnorman/miniWeather

## 2 Mathematical Formulation and Numerical Discretization

### 2.1 Moist, Compressible Large Eddy Simulation Equations

The atmospheric Large Eddy Simulation (LES) formulation is based on the moist, compressible, filtered Navier-Stokes equations of gas dynamics with non-hydrostatic buoyancy-driven motions. The gas is treated as a sum of ideal gases, dry air and water vapor – with immersed hydrometeors that contribute to mass but not to pressure. The sub-grid-scale dissipation is assumed to be dominated by eddies, and this model uses a common eddy viscosity formulation Lilly (1966, 1967). The equations are cast in Cartesian geometry as follows:

$$\frac{\partial}{\partial t}\begin{bmatrix} \rho \\ \rho u \\ \rho v \\ \rho w \\ \rho\theta \\ \rho q_v \\ \rho K \\ \rho q_\ell \\ \rho q_\mathcal{P} \end{bmatrix} + \frac{\partial}{\partial x}\begin{bmatrix} \rho u \\ \rho uu + p' + \tau_{11} \\ \rho uv + \tau_{21} \\ \rho uw + \tau_{31} \\ \rho u\theta + \tau_{\theta 1} \\ \rho uq_v + \tau_{v1} \\ \rho uK + \tau_{K1} \\ \rho uq_\ell + \tau_{\ell 1} \\ \rho uq_\mathcal{P} + \tau_{\mathcal{P}1} \end{bmatrix} + \frac{\partial}{\partial y}\begin{bmatrix} \rho v \\ \rho vu + \tau_{12} \\ \rho vv + p' + \tau_{22} \\ \rho vw + \tau_{32} \\ \rho v\theta + \tau_{\theta 2} \\ \rho vq_v + \tau_{v2} \\ \rho vK + \tau_{K2} \\ \rho vq_\ell + \tau_{\ell 2} \\ \rho vq_\mathcal{P} + \tau_{\mathcal{P}2} \end{bmatrix} + \frac{\partial}{\partial z}\begin{bmatrix} \rho w \\ \rho wu + \tau_{13} \\ \rho wv + \tau_{23} \\ \rho ww + p' + \tau_{33} \\ \rho w\theta + \tau_{\theta 3} \\ \rho wq_v + \tau_{v3} \\ \rho wK + \tau_{K3} \\ \rho wq_\ell + \tau_{\ell 3} \\ \rho wq_\mathcal{P} + \tau_{\mathcal{P}3} \end{bmatrix} = \begin{bmatrix} 0 \\ 0 \\ 0 \\ -\rho'g \\ 0 \\ 0 \\ K_S + K_D + K_B \\ 0 \\ 0 \end{bmatrix} \tag{1}$$

$$p = (\rho_d R_d + \rho_v R_v)T = \rho R^\star T = C_0 (\rho\theta)^\gamma; \qquad \frac{\theta}{T} = \frac{R^\star}{R_d}\left(\frac{p_0}{p}\right)^{R_d/c_p}; \qquad R^\star = \frac{\rho_d}{\rho}R_d + \frac{\rho_v}{\rho}R_v \tag{2}$$

where $\rho$ is total density; $u$, $v$, and $w$ are wind velocities in the $x$, $y$, and $z$ directions, respectively; $\theta$ is a form of virtual potential temperature defined by equations (2); $q_v$ is the wet water vapor mixing ratio (such that $\rho_v = \rho q_v$ is the density of water vapor); $K$ is unresolved, sub-grid-scale specific Turbulence Kinetic Energy (TKE); $q_\ell$ is a tracer quantity that contributes to mass but not to pressure; $q_\mathcal{P}$ is a passive tracer that is mass-weighted but contributes to neither mass nor pressure (e.g., mass-weighted number concentration for two-moment microphysics); $p$ is total ideal gas pressure defined by equation (2), a sum of dry and water vapor pressures; $C_0 = \left(R_d (p_0)^{-R_d/c_p}\right)^\gamma$ is a constant of proportionality; $\gamma = c_p/c_v$ is the ratio of specific heats of dry air; $c_p$ is specific heat of dry air at constant pressure; $c_v$ is specific heat of dry air at constant volume; $p_0$ is reference pressure; $R_d$ is the dry air ideal gas constant; $R_v$ is the water vapor ideal gas constant; $K_S$ is TKE shear production; $K_D$ is TKE dissipation; $K_B$ is the TKE buoyancy source/sink; $\tau_{ij}\ \forall i,j \in \{1,2,3\}$ is the unresolved eddy flux of wind momenta; $\tau_{\theta j}\ \forall j \in \{1,2,3\}$ is eddy flux of potential temperature; $\tau_{vj}$ is eddy flux of water vapor; $\tau_{Kj}$ is eddy flux of TKE (i.e., the "turbulence transport" terms for TKE); $\tau_{\ell j}$ is the eddy flux of tracers that contribute to mass but not to pressure (e.g., hydrometeors); $\tau_{\mathcal{P}j}$ is the eddy flux of passive mass-weighted tracers contributing to neither mass nor pressure; and $g$ is acceleration due to gravity. The total density, $\rho$, is defined as the sum of all mass-contributing densities: $\rho = \rho_d + \rho_v + \sum_\ell \rho_\ell$, where $\rho_d$ is the density of dry air. The quantities in equation (1) are filtered quantities.

The perturbation pressure, $p' = p - p_H$, and perturbation density, $\rho' = \rho - \rho_H$, are deviations from a dominant hydrostatic balance denoted by:

$$\frac{dp_H}{dz} = -\rho_H g \tag{3}$$

which can be defined arbitrarily. In these simulations, the balance is obtained with either a vertical potential temperature profile or a combination of a vertical temperature profile and a vertical water vapor dry mixing ratio profile. The purpose of removing hydrostasis is to better resolve the perturbations, which form the primary means of forcing for buoyancy-driven atmospheric flow. Poor resolution of hydrostasis results in spurious vertical velocities, which can render the solution physically unrealizable.

Equation set (1) will also be expressed in places using a generic vector form of conservation laws:

$$\frac{\partial \mathbf{q}}{\partial t} + \frac{\partial \mathbf{f}}{\partial x} + \frac{\partial \mathbf{g}}{\partial y} + \frac{\partial \mathbf{h}}{\partial z} = \mathbf{s} \tag{4}$$

### 2.1.1 Eddy Fluxes

In eddy viscosity models, particularly for the atmosphere, the molecular viscosity is so small as to be practically negligible in its direct dissipative influence. However, it leads to a cascade of eddies (only some of which are resolved) that eventually lead to dissipation at the larger scales. The basis of eddy viscosity models is to break the flow into resolved scales and unresolved perturbations. In this study, an "implicitly filtered" eddy viscosity model is used, meaning cell-averaged quantities from the Finite-Volume space-time discretization are considered to be resolved and all perturbations below the grid scale are considered unresolved.

Using averaging rules, the only quantities that remain after integration are products of resolved winds, $\overline{u}_i \overline{u}_j$, and averages of unresolved perturbation products such as $\overline{u'_i u'_j}$, or correlations. We employ the commonly used gradient diffusion assumption, assuming isotropically scaled eddies in the three spatial directions. The eddy viscosity, $K_m$, is considered to be a function of unresolved TKE and a stability-corrected length scale, giving an eddy flux of:

$$\tau_{ij} = -\rho(K_m + \nu)\left(\frac{\partial u_i}{\partial x_j} + \frac{\partial u_j}{\partial x_i} - \frac{2}{3}\frac{\partial u_k}{\partial x_k}\delta_{ij}\right); \qquad \tau_{\theta j} = -\rho\left(\frac{K_m}{Pr_T} + \frac{\nu}{Pr}\right)\frac{\partial \theta}{\partial x_j} \tag{5}$$

$$\tau_{Kj} = -2\rho(K_m + \nu)\left(\frac{\partial K}{\partial x_j}\right); \qquad \tau_{\ell j} = -2\rho\left(\frac{K_m}{Pr_T} + \frac{\nu}{Pr}\right)\left(\frac{\partial \rho_\ell}{\partial x_j}\right); \qquad K_m = 0.1L\sqrt{K} \tag{6}$$

$$Pr_T = \frac{\Delta}{1 + 2L}; \qquad L = \min\left(\frac{0.76\sqrt{K}}{N + \epsilon}, \Delta\right); \qquad \Delta = (\Delta x \Delta y \Delta z)^{1/3}; \qquad N = \begin{cases} \sqrt{(g/\theta)(\partial\theta/\partial z)} & \text{if } \partial\theta/\partial z > 0 \\ 0 & \text{otherwise} \end{cases} \tag{7}$$

where $\epsilon = 10^{-20}$ is a small number to avoid division by zero; $N$ is the Brunt–Väisälä frequency, a measure of atmospheric stability; $Pr_T$ is the turbulent Prandtl number; $Pr$ is the Prandtl number; $\nu$ is kinematic viscosity; $\delta_{ij}$ is the Kronecker delta; $u_1$, $u_2$, and $u_3$ are the velocity components in the $x$, $y$, and $z$ directions (i.e., $u$, $v$, and $w$); and $x_1$, $x_2$, and $x_3$ are the $x$, $y$, and $z$ directions. In equation (5), the $k$ indices below are summed over the three indices for $k$ (Einstein summation is implied). For the derivatives, second-order-accurate finite-differences valid at the appropriate cell faces for each individual flux are used.

### 2.1.2 TKE Sources and Sinks

The TKE evolution equation contains advection and turbulent transport (diffusion) on the left-hand-side of equation (1). The remaining processes resolved here are the sources and sinks that are, in general, not cast in conservation form. They are defined as:

$$K_S = \sum_{i,j \in \{1,2,3\}} \frac{\rho K_m}{2} \left( \frac{\partial u_i}{\partial x_j} + \frac{\partial u_j}{\partial x_i} \right) \left( \frac{\partial u_i}{\partial x_j} + \frac{\partial u_j}{\partial x_i} \right); \quad K_D = -\rho \left( 0.19 + 0.51 L \Delta^{-1} \right) \Delta^{-1} K^{3/2}; \quad K_B = -\frac{\rho g K_m}{\theta (Pr)} \frac{\partial \theta}{\partial z}$$

$$(8)$$

### 2.1.3 Surface Fluxes

Surface fluxes at the model surface and at immersed boundaries are enforced with Monin–Obukhov similarity theory (Monin and Obukhov, 1954). For all surface cells, and for all cells adjacent to an immersed cell, surface fluxes in each direction are applied via the following flux terms in a corresponding manner as the eddy fluxes in equation (1):

$$\frac{\partial}{\partial t} \begin{bmatrix} u \\ v \\ w \\ T \end{bmatrix} + \frac{\partial}{\partial x} \begin{bmatrix} 0 \\ \tau_{21}^* \\ \tau_{31}^* \\ \tau_{T1}^* \end{bmatrix} + \frac{\partial}{\partial y} \begin{bmatrix} \tau_{12}^* \\ 0 \\ \tau_{32}^* \\ \tau_{T2}^* \end{bmatrix} + \frac{\partial}{\partial z} \begin{bmatrix} \tau_{13}^* \\ \tau_{23}^* \\ 0 \\ \tau_{T3}^* \end{bmatrix} = \begin{bmatrix} 0 \\ 0 \\ 0 \\ 0 \end{bmatrix} \tag{9}$$

$$\tau_{21}^\star = -\frac{\kappa^2 v \sqrt{v^2 + w^2}}{\xi^2 \Delta x} \;\; ; \;\; \tau_{31}^\star = -\frac{\kappa^2 w \sqrt{v^2 + w^2}}{\xi^2 \Delta x} \;\; ; \;\; \tau_{T1}^\star = -\frac{\kappa^2 (T - T_I) \sqrt{v^2 + w^2}}{\xi^2 \Delta x} \tag{10}$$

$$\tau_{12}^\star = -\frac{\kappa^2 u \sqrt{u^2 + w^2}}{\eta^2 \Delta y} \;\; ; \;\; \tau_{32}^\star = -\frac{\kappa^2 w \sqrt{u^2 + w^2}}{\eta^2 \Delta y} \;\; ; \;\; \tau_{T2}^\star = -\frac{\kappa^2 (T - T_I) \sqrt{u^2 + w^2}}{\eta^2 \Delta y} \tag{11}$$

$$\tau_{13}^\star = -\frac{\kappa^2 u \sqrt{u^2 + v^2}}{\zeta^2 \Delta z} \;\; ; \;\; \tau_{23}^\star = -\frac{\kappa^2 v \sqrt{u^2 + v^2}}{\zeta^2 \Delta z} \;\; ; \;\; \tau_{T3}^\star = -\frac{\kappa^2 (T - T_I) \sqrt{u^2 + v^2}}{\zeta^2 \Delta z} \tag{12}$$

$$\xi = \ln \left( \frac{\frac{\Delta x}{2} + z_0}{z_0} \right) \;\; ; \;\; \eta = \ln \left( \frac{\frac{\Delta y}{2} + z_0}{z_0} \right) \;\; ; \;\; \zeta = \ln \left( \frac{\frac{\Delta z}{2} + z_0}{z_0} \right) \tag{13}$$

where $z_0$ is the roughness length, and $T_I$ is the temperature of the immersed cell. If a temperature is omitted, no forcing toward temperature is applied. One can also implement a heat flux, $\overline{u'\theta'}$ (analogous for $y$ and $z$ directions), which is added as a flux from the material surface. Fluxes are applied to wind velocity and to temperature because this is implemented as a separate module in portUrb and operates directly on the coupler state. Since immersed material is forced toward the hydrostatic background potential temperature, thermally, the absolute temperature value, $T$, of immersed material will generally change with height for stratified flows when no direct temperature is specified.

## 2.2 Dimensionally Split Finite-Volume Discretization

Using the vector form of the LES equations in equation (4), we split the equations such that each dimension is handled separately, and the vector form equations become, in general:

$$\frac{\partial \mathbf{q}}{\partial t} + \frac{\partial \mathbf{f}}{\partial x} = \mathbf{s} \tag{14}$$

This is integrated in space (semi-discretized) over domain-spanning cells of equal spacing, $\Omega_i = \left[ x_{i-1/2}, x_{i+1/2} \right]$, where $x_{i\pm 1/2} = x_i \pm \Delta x/2$. The divergence theorem is applied, and an upwind Riemann solver is used to give the form:

$$\frac{\partial \overline{\mathbf{q}}_i}{\partial t} + \frac{\mathcal{R}\left( \widetilde{\mathbf{q}}_i \left( x_{i+1/2} \right), \widetilde{\mathbf{q}}_{i+1} \left( x_{i+1/2} \right) \right) - \mathcal{R}\left( \widetilde{\mathbf{q}}_{i-1} \left( x_{i-1/2} \right), \widetilde{\mathbf{q}}_i \left( x_{i-1/2} \right) \right)}{\Delta x} = \mathbf{s}\left( \overline{\mathbf{q}}_i \right) \tag{15}$$

where $\widetilde{\mathbf{q}}_i (x)$ is a reconstruction of the variation of $\mathbf{q}$ within the cell of domain $\Omega_i$, $\mathcal{R}\left( \mathbf{q}^-, \mathbf{q}^+ \right)$ is a Riemann solver that returns a flux vector, $\mathbf{f}(\mathbf{q})$, from multi-valued states at cell edges using the upwind state based on locally frozen characteristics (the flux Jacobian diagonalization: $\partial \mathbf{f}/\partial \mathbf{q} = L^{-1} \Lambda L$). The state $\left( \mathbf{q}^- + \mathbf{q}^+ \right)/2$ is used to compute the diagonalization for upwinding. For simplicity, a constant acoustic speed of 350 m/s is assumed, and supersonic flow is not supported in the Riemann solver. This is the same approach used in Norman et al. (2023b).

While the gravity and eddy viscosity source term can be integrated with high-order discretizations, the hyperbolic dynamics use constant and uniform gravity, and the diffusive SGS dynamics do not benefit substantially from high-order accuracy. For the reconstruction, $\widetilde{\mathbf{q}}_i (x)$, eleventh-order Weighted Essentially Non-Oscillatory (WENO) limiting (Liu et al., 1994) is used along with weight mapping (Feng et al., 2012). When a cell is within ten cells of an immersed boundary in any direction, weight mapping is omitted to reduce oscillations because immersed boundaries essentially represent permanent contact discontinuities.

## 2.3 Strong Stability Preserving Runge-Kutta Time Discretization

The optimal, three-stage, third-order accurate Strong Stability Preserving Runge-Kutta (SSPRK3) time discretization is used to integrate the remaining temporal Ordinary Differential Equation resulting from the semi-discretization in section 2.2. The authors experimented with different dimensional splitting approaches, and it was found that the least numerically diffused results came from splitting the dimsensions fully independently within each stage of the SSPRK3 time stepping. While this limited the maximum stable CFL value to roughly 0.7, it resulted in less numerical dissipation than using dimensional splitting techniques outside the SSPRK3 time stepping procedure. This approach with SSPRK3 also produced less numerical diffusion than attempts at using an ADER time discretization with Differential Transforms (Norman and Finkel, 2012; Norman, 2013a, 2021) with dimensional splitting outside the ADER computations. This approach to dimensional splitting within each RK stage has also been shown to reduce errors at singularities on non-orthogonal grids such as the cubed-sphere (Katta et al., 2015).

## 2.4 Free-Slip Immersed Boundaries with Surface Friction

### 2.4.1 Modifying Variables on the Grid

Simplified immersed boundaries are implemented to represent solid surfaces embedded in the flow. These only represent the proportion of immersed material in a cell along with the roughness length of that immersed material as well as the temperature and / or heat flux. They are implemented similarly to how the solid wall boundaries are implemented at the bottom and top of the domain. Each cell will have an immersed "proportion", $\overline{\sigma}_{i,j,k} \in [0,1]$, which is the proportion of the cell that is immersed material. Before each Runge-Kutta stage, immersed boundaries are nudged toward hydrostatic values at rest according to the fifth power of the immersed proportion and a timescale, $\mathcal{T}_{i,j,k} \in [1, \infty)$:

$$\frac{d\overline{\mathbf{q}}_{i,j,k}}{dt} = \frac{(\overline{\sigma}_{i,j,k})^5}{\mathcal{T}_{i,j,k}} \left( \overline{\mathbf{q}}_{H,k} - \overline{\mathbf{q}}_{i,j,k} \right) \tag{16}$$

where $\overline{\mathbf{q}}_{H,k} = \left[ \overline{\rho}_{H,k}, 0, 0, 0, \overline{\rho\theta}_{H,k}, 0, 0, 0 \right]^\top$ and $\mathcal{T}_{i,j,k} = 1$ means the cell is immediately set to its target value. $\mathcal{T}_{i,j,k}$ is, in general, the time scale in terms of number of time steps for adjustment to a hydrostatic state at rest. The fifth power is obtained from numerical experiments, matching low resolution to high resolution solutions for a coarsely resolved sphere in section 3.5.

The timescale, $\mathcal{T}_{i,j,k}$, is 8, 4, and 2 if the cell is immersed and is within one, two, or three cells in all directions (including corners) of a non-immersed boundary. All other immersed cells have a timescale of 1, meaning they are immediately set between every stage to the target immersed values. Essentially, this approach "softens" the immersed discontinuity slightly when the flow switches from non-immersed to immersed to reduce Runge oscillations in the presence of high-order reconstruction. This approach also ensures that all immersed boundaries get close to their targets each time step and that internal regions of larger immersed structures like buildings do not allow any flow through at all.

In general, when the user ingests building or terrain geometry, it can be quite sharp in its features. The use of WENO limiting, the avoidance of WENO weight mapping near immersed material, the use of an upwind Riemann solver, and the slight softening of immersed forcing time scale near the outside of structures appears to be enough to avoid visible Runge oscillations while simultaneously striving for a large amount of resolved turbulence and boundary resolution per degree of freedom.

### 2.4.2 Special Treatment for Tangential Winds and Pressure

In a given direction, the normal wind velocity boundary condition is zero inside immersed boundaries. The tangential winds, however, need to be free-slip so that surface friction can later be applied to impose a roughness length on the tangential flow. To enable this, for transverse velocities relative to a given direction (recall the dynamics are implemented in a dimensionally split fashion), the moment an immersed proportion greater than zero is encountered, the value of the last non-immersed cell before an immersed boundary is replicated throughout the rest of that direction in a stencil. This, in effect, implements a zero derivative Neumann boundary for tangential wind velocities to create a free-slip solid wall boundary. This way, normal velocity is forced toward zero, but tangential velocity is not. This implementation is ad hoc to a dimensionally split implementation.

In this model, during the semi-discretized spatial discretization of the dynamics, pressure is treated as its own separate reconstructed variable. Pressure is given a zero derivative boundary condition with the same treatment used for the two tangential wind velocities in a given direction.

### 2.4.3 Modifications to LES Closure and Surface Friction

Many of the modules in the model need to be changed to exhibit physical behavior in the presence of immersed boundaries, where mass and thermodynamics variables are set to hydrostatic values, and wind velocities and tracers are set to zero. The LES closure, for instance, cannot mix tangential zero velocity values in immersed cells with adjacent non-immersed cells, or it would essentially imply a surface friction term without consideration of roughness length or Monin-Obukhov similarity theory, which is in the territory of the surface friction scheme, not the LES closure. Therefore, the LES SGS closure sets SGS interface

fluxes, $\tau$, to zero whenever one of the two neighboring cells is immersed.

The surface friction scheme is modified to no longer only work at the surface but to impose fluxes on the tangential wind and temperature at any interface where one of the two adjacent cells is immersed. For already immersed cells, this has little to no effect because the cell is already being forced toward immersed target values (hydrostatic flow at rest) in the dynamical core. For non-immersed cells adjacent to an immersed cell, this imposes friction and thermal forcing according to immersed

temperature, roughness length, or heat flux. Those three values along with the immersed proportion are defined at every cell in the domain so that one can customize the roughness, temperature, and heat flux of any immersed material in the domain. If one desires no-slip immersed boundaries, the immersed roughness can be set to a large value. If one desires fully free-slip, it can be set to a small value.

### 2.4.4 Modifications to the WENO Reconstruction Approach

WENO reconstruction performs reconstruction over multiple sub-stencils within an overall stencil and uses the weighted sum of reconstructions. The higher the Total Variation (a measure of oscillation) of a stencil, the lower its weight. Particularly for normal velocity in a given direction, the zero value in boundary ghost cells lead to all stencils containing those ghost cells having a permanent discontinuity. Thus, WENO limiting weights the stencil without ghost cells quite highly and essentially ignores the boundary conditions.

It is not immediately clear how this leads to artifacts in the solution, but two options are the most likely culprits. First, when this behavior occurs, boundary cells end up using fully one-sided reconstructions most of the time, which have lower stability than reconstructions containing cells on both sides of the reconstructed cell (Norman, 2021). Second, when the edge flux is zero, but the reconstruction has no notion of the boundary conditions, this can lead to one cell face with large mass flux and the other cell face with zero mass flux. This can lead to creating higher pressures and, in turn, solution artifacts in cells adjacent to

boundaries.

To avoid both of these possibilities, we alter the Total Variation (TV) of the stencil that *does not* contain the boundary ghost cells to have the maximum TV of the other stencils. This way, the stencil without ghost cells cannot consume the majority of the weighted sum of stencils, and at least one of the other stencils containing ghost cell information will have influence on the

final WENO polynomial, injecting boundary information to the final reconstruction and ensuring the reconstruction is not fully
one-sided most of the time. It also allows the WENO weighting procedure to generally choose the stencils with lowest TV –
an essential component of non-oscillatory reconstruction.

## 2.5   Variable Vertical Grid Spacing

The vertical grid in portUrb can be arbitrarily set by providing a set of monotonically increasing vertical interface levels. This
is supported simply in the numerics by use of a metric transformation from the physical vertical coordinate, $z$, to an equally
spaced reference coordinate: $\zeta(z)$. For convenience, the $\zeta$ coordinate is simply set to be the vertical interface index such that
the grid spacing in $\zeta$ coordinates is always one. Before reconstruction, the cell average, $\overline{q}_k$, is multiplied by the physical grid
spacing, $\Delta z_k$, to transform into $\zeta$ coordinates (recalling that $\Delta \zeta \equiv 1$). Then, after reconstruction in $\zeta$ coordinates, the cell-edge
values are multiplied by the inverse of a high-order reconstruction of $dz/d\zeta$ at the cell edge in question to transform back into
physical vertical coordinates.

The portUrb code has several convenience functions to generate a stretched vertical grid. There is a function to generate
equal grid spacing. There is a function that generates exponentially increasing vertical grid spacing based on an input of the
vertical domain extent and the desired number of vertical levels. The routine fits the constant multiple by which grid spacing
increases between successive levels to those parameters. There is also a function that takes an input of a lower level grid spacing
and a higher level grid spacing along with the vertical domain extent and the domain over which the grid spacing transitions
between the two using smooth polynomial interpolation. This routine determines the number of vertical grid cells rather than
having that value specified by the user.

## 2.6   Boundary Conditions

There are four different boundary conditions that can be specified in the dynamical core and SGS TKE closure routines:
periodic, solid wall, open, and precursor forcing. The precursor boundary forcing is implemented by declaring one coupler
object to be the concurrent precursor simulation. That simulation then saves the state in the domain's "ghost cells" for use by
the forced simulation for each of the Runge-Kutta stages. Then, the saved ghost cells from the precursor simulation are copied
into the forced simulation's ghost cells at each Runge-Kutta stage if the velocity normal to the boundary is flowing into the
domain. An open boundary is used if the velocity normal to the boundary is flowing out of the domain.

## 2.7   Model Construction

The model is constructed with a "coupler" at the core. The coupled state is defined to be dry air density, wind velocities in
all three directions, absolute temperature, and tracer densities. For each tracer, it is defined whether the tracer should remain
positive, whether it contributes to mass, and whether it is diffused with the SGS closure. The only required tracers are water
vapor density (for all simulations) and mass-weighted TKE (if the LES closure module is used). All tracers must be specified
as mass-weighted or total density. This coupled state has the benefit of being non-ambiguous (whereas definitions of potential

temperature and mixing ratios can be varied). A disadvantage computationally is that most parameterizations and dynamical cores use mixing ratios and potential temperature, meaning a conversion is needed before and after every module, typically. This cost is deemed acceptable to have a non-ambiguous coupler state.

A "module" is a self-contained action that changes the coupled state or produces output, and all model actions are implemented as modules. The coupler state is passed to a module along with the desired time step. Modules may sub-step to maintain stability as needed, but in the experiments in this study, the dynamical core imposes the smallest time step, and all modules are run at the dynamical core time step for now. The initial state, therefore, is implemented based on the coupler's state rather than the dynamical core state defined in equation (1).

## 2.8 Ensembles

Ensembles within a single executable are flexibly supported with a core module called an "Ensembler", a class that manages ensembles. One can implement different "dimensions" of ensembles, specify how many tasks (relative to the smallest ensemble member) a given member needs, split the overall MPI communicator into sub-communicators (one for each ensemble member), and redirect the output of each member to its own file. This way, each ensemble member can do any arbitrary operation, read from any individual file, and output to any arbitrary file. With this approach, one can submit capability-class jobs (that is, jobs that use 20% or more of the machine at a time) on supercomputers such as DOE Leadership Computing Facilities like the Oak Ridge Leadership Computing Facility or the Argonne Leadership Computing Facility. This capability is used in several of the numerical experiments in section 3.

## 2.9 Portable C++ Approach

The model is coded in a portable C++ library called Yet Another Kernel Launcher (YAKL) (Norman et al., 2023a), which is based on the Kokkos portable C++ library (Trott et al., 2021). The core of these libraries is to wrap kernel code (ostensibly the code inside a set of loops over the grid cells being operated on) in a class object. That object is then passed to a launcher that launches the code in parallel on a given architecture. For Nvidia, AMD, and Intel GPUs, the CUDA, HIP, and SYCL specifications are used to launch the code, respectively. There are other backends available as well such as OpenMP threading for multi-core CPUs. All modules in portUrb that operate on a set of grid cells are run on whatever "device" is available on a given machine (typically a GPU). Care has been taken to ensure all kernels run efficiently by streaming memory effectively from DRAM (ensuring higher GPU memory bandwidth is realized) (Norman, 2013b) and keeping kernel resource usage low enough to use the entire GPU effectively (a concept known as "occupancy") (Shobaki et al., 2020).

## 3 Numerical Experiments

The initial states for all experiments are implemented with 9-point Gauss Legendre Lobatto (GLL) quadrature in all directions within each cell. All experiments use a CFL value of 0.6 assuming a maximum wave speed of 450m/s (roughly 350 m/s acoustic speed at the surface with up to 100 m/s wind speeds). All experiments use equal grid spacings in all directions. An initial TKE

of $10^{-6}$ is initialized uniformly to seed initial unresolved TKE. This initial value appears to have little effect on the solution as it quickly resolves into an equilibrium state driven largely by pressure gradient or geostrophic forcing, shear production at immersed interfaces, and by buoyancy in some simulations.

## 3.1 Neutral Atmospheric Boundary Layer

### 3.1.1 Specification

This experiment simulates a buoyantly neutral atmospheric boundary layer seeded by initial temperature perturbations and maintained by geostrophic forcing (described in section 3.1.2). A domain of $4 \times 4 \times 1$ km is used to simulate for ten model hours with 10 m grid spacing. This test case uses an initial potential temperature profile of:

$$\theta_{initial}(z) = \begin{cases} 300 & \text{if } z < 500\text{m} \\ 300 + 0.08\,(z - 500) & \text{if } z \geq 500\text{m and } z < 650 \\ 300 + 0.08\,(150) + 0.003(z - 650) & \text{if } z \geq 650\text{m} \end{cases} \tag{17}$$

This creates a neutral environment in the lowest 500m, a strong stable inversion between 500m and 650m, and a weaker inversion above 650m. An initial wind velocity of $u = 10$ m s$^{-1}$ and $v = w = 0$ is specified. From this, an initial surface pressure of $10^5$ Pa along with hydrostasis and the equation of state define a hydrostatically balanced initial state that provides the density. Finally, cell-wise random uniform perturbations in the domain $[-0.25, 0.25]$ K are added to the temperature field. A surface roughness length of 0.1m is used. Geostrophic forcing is used with $u_G = 10$ m s$^{-1}$, $v_G = 0$ m s$^{-1}$, and $\phi_G \approx 43.289°$ (the exact value is the solution to $2\Omega \sin \phi_G = 10^{-4}$) as described in section 3.1.2.

This uses no surface heat flux, and the surface temperature is set to 300K. Periodic boundary conditions are used in the horizontal directions, and solid no-slip solid walls are used in the vertical direction. Surface fluxes are only applied at the bottom boundary, and a sponge layer is implemented in the top 10% of the model with a forcing scaling of $z^3$ to force the model variables to zero for everything except horizontal velocities, which are forced to the horizontal mean values at each respective vertical level.

### 3.1.2 Geostrophic Forcing

The model is forced to an equilibrium state with geostrophic forcing, which imposes the following extra source term to the horizontal wind velocities:

$$\frac{\partial}{\partial t} \begin{bmatrix} \overline{u}_{i,j,k} \\ \overline{v}_{i,j,k} \end{bmatrix} = \begin{bmatrix} 2\Omega \sin \phi_G \left( \overline{v}_{k,horiz} - v_G \right) \\ -2\Omega \sin \phi_G \left( \overline{u}_{k,horiz} - u_G \right) \end{bmatrix} \tag{18}$$

In equation (18), $\overline{u}_{k,horiz}$ and $\overline{v}_{k,horiz}$ are the averages of $u$ and $v$ over the global horizontal domain at the vertical level, $k$. Equation set (18) in continuous form as it would be applied to equation set (1) would appear as:

$$\frac{\partial}{\partial t} \begin{bmatrix} \rho u \\ \rho v \end{bmatrix} = \begin{bmatrix} 2\rho\Omega \sin \phi_G \left( \langle v \rangle_{horiz} - v_G \right) \\ -2\rho\Omega \sin \phi_G \left( \langle u \rangle_{horiz} - u_G \right) \end{bmatrix}, \tag{19}$$

where $\langle u \rangle_{horiz}$ and $\langle v \rangle_{horiz}$ (both functions of $z$ and time) are the horizontal averages of $u(x,y,z,t)$ and $v(x,y,z,t)$, respectively.

### 3.1.3 Results

Figure 1 plots the wind magnitude for the neutral atmospheric boundary layer test case at ten model hours. The scale and angle of turbulent fluctuations as well as behavior near the inversion match that of Figure 1 of Sauer and Muñoz-Esparza (2020). The supergeostrophic wind speed at the inversion is present as well (Pedersen et al., 2014). Figure 2 plots vertical profiles of the domain mean $u$ and $v$ velocities as a function of height at 8 and 10 model hours. Also plotted are the vertical domain mean profiles from Figure 2 of Sauer and Muñoz-Esparza (2020) as comparison points. Both models show the supergeostrophic wind at the inversion and increase in wind speed from 8 to 10 model hours in both horizontal wind directions. However, the $u$-direction wind speed in portUrb is lower aloft and higher at the surface. The $v$-velocity is quite similar between portUrb and FastEddy at $t = 10$hr, but portUrb has a smaller difference between 8 and 10 hours of simulation. The potential temperature profile is also similar, showing a slight growth in the inversion height and smoothing of the initial inversion temperature discontinuity. The second potential temperature discontinuity at $z = 650$m is less diffused in portUrb than it is in FastEddy. The magnitudes of $u$-velocity and $v$-velocity are slightly lower in this study, possibly due to the use of a larger grid spacing, but the structure remains similar.

Figure 3 shows plots of mean values, minimum to maximum ranges, and the first and third quartiles for wind speed, $\overline{u'w'}$ correlation, and total Turbulence Kinetic Energy (TKE). Also plotted are the mean data from Figure 3 in Sauer and Muñoz-Esparza (2020) for comparison. A sharp peak in TKE is observed near the surface, decaying toward zero aloft. The TKE for portUrb is in quite close agreement with FastEddy. A log profile of wind speed increase from the frictioned bottom boundary is shown in Figure ?? using reference height of 500m and a reference wind speed of 10 m/s for the log-law reference. The horizontal mean resolved $\overline{u'w'}$ correlation reaches a slightly larger peak magnitude in this study and does so more smoothly than observed in Sauer and Muñoz-Esparza (2020), possibly due to the use of WENO limiting, a fully upwind flux, or a coarser grid spacing. A plot of the wind speed spectra averaged among 1-D FFTs in the two horizontal directions at a height of $z = 75$m are given in Figure 4, showing an effective resolution of about $8\Delta x$ and perhaps slightly higher. No accumulation of energy is observed at $2\Delta x$ scales, demonstrating physically realizable dissipation of energy at the end of the 3-D cascade by the numerical discretization and SGS turbulence closure.

## 3.2 Dry Convective Boundary Layer

### 3.2.1 Specification

This experiment simulates a dry convective boundary layer with an initially neutral boundary layer up to 600m, an inversion above 600m, and a positive heat flux at the model surface to mimic radiative heating emanating from the model surface. The

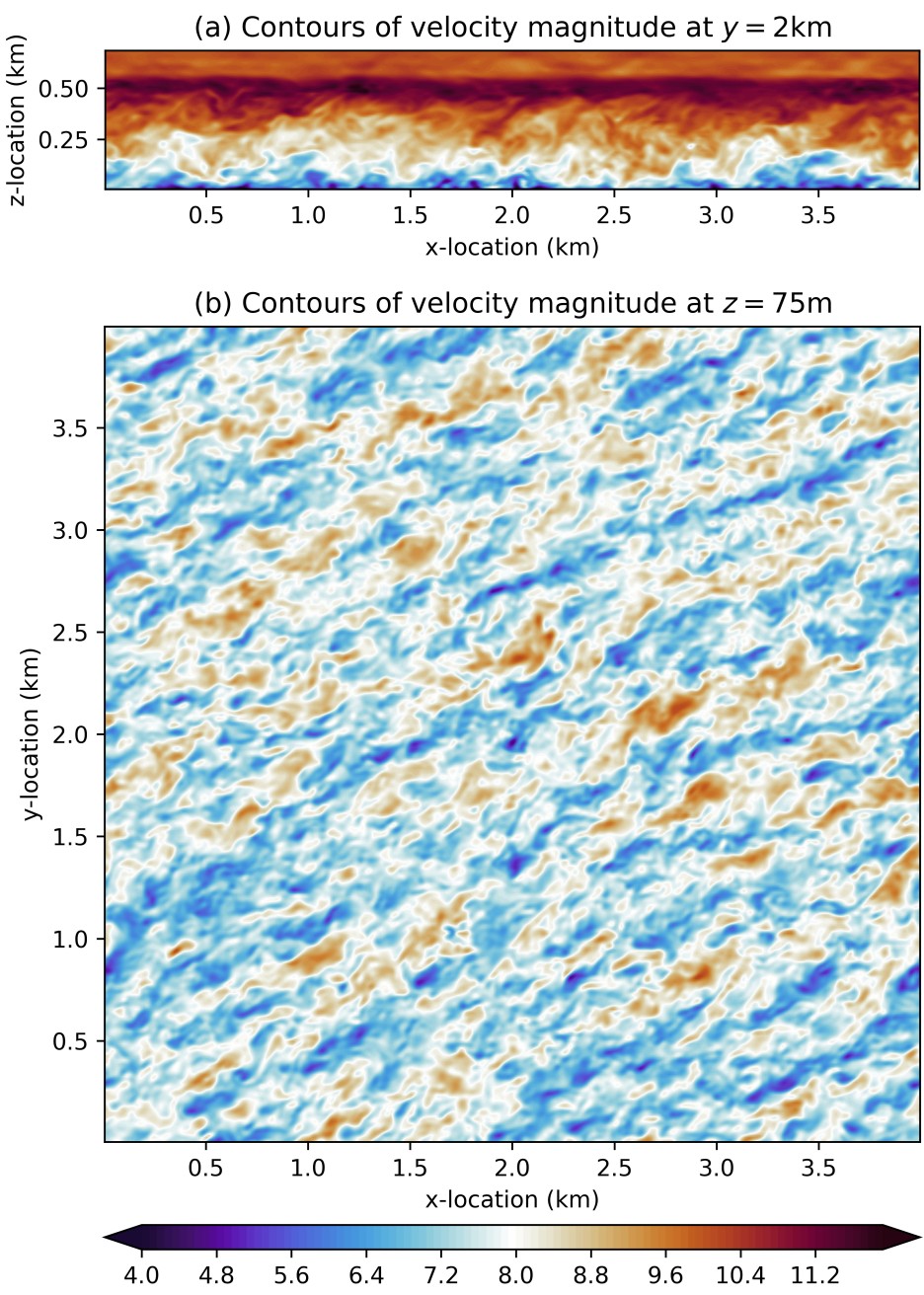

**Figure 1.** Plots of wind magnitude in m/s for the neutral ABL test case at ten model hours.

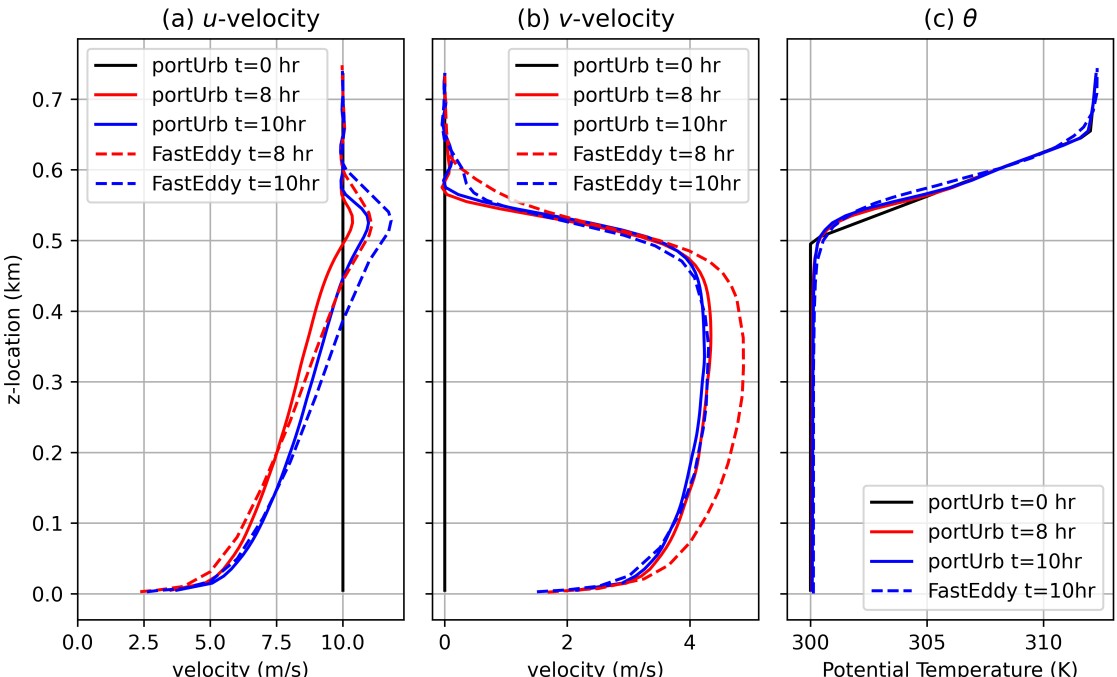

**Figure 2.** Plots for the neutral ABL test case of horizontally-averaged $u$-velocity (m/s), $v$-velocity (m/s), and virtual potential temperature (K) at different times.

potential temperature is initialized as:

$$\theta_{initial}(z) = \begin{cases} 309 & \text{if } z < 600\text{m} \\ 309 + 0.004\,(z - 600) & \text{if } z \geq 600\text{m} \end{cases} \tag{20}$$

This is simulated over a domain of $6 \times 6 \times 3$ km for a time of three model hours with 10m grid spacing. The surface temperature is set to 309K, but it is not continuously enforced via surface fluxes. Rather, a heat flux of $0.4$ K m s$^{-1}$ is imposed at the bottom boundary to initiate convection. Wind is initialized as $u = 10$ m/s and $v = w = 0$. A surface roughness length of 0.05m is used, and geostrophic forcing is applied with $u_G = 10$, $v_G = 0$, and $\phi_G = 33.5°$. Random uniform temperature perturbations are added to the lowest 400m of the domain in the range of $[-0.25, 0.25]$K in the initial state. Periodic boundary conditions are

used in the horizontal, and solid no-slip walls are used in the vertical. Surface fluxes are only applied at the bottom boundary, and the same sponge layer as used in section 3.1 is used in the top 10% of this simulation as well.

### 3.2.2   Results

Figure 5 plots the potential temperature and vertical velocity at $z = 75$m as well as $y = 3$km at a time of $t = 2$hr. Comparing this against Figure 8 of Sauer and Muñoz-Esparza (2020) and Figure 3 of Mirocha et al. (2018), the scale of structures are similar.

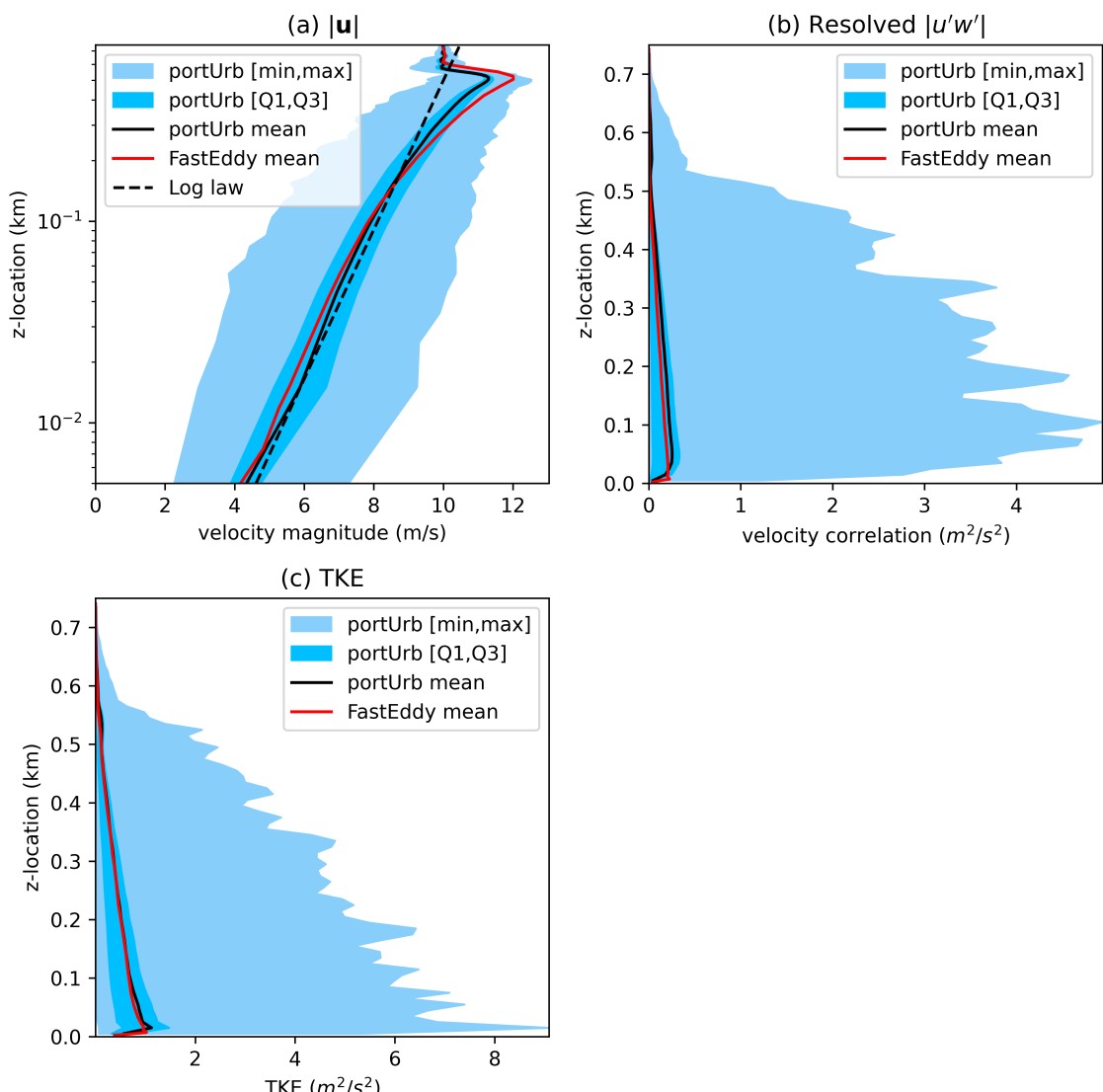

**Figure 3.** Plots for the neutral ABL test case at ten model hours of the mean (black line), first to third quartile range (dark blue shading), and minimum to maximum range (light blue shading) for various quantities. Perturbation quantities in this plot are deviations from the horizontal mean at a given vertical level. Resolved TKE is defined as $\overline{\rho(u')^2 + (v')^2 + (w')^2}$, and unresolved SGS TKE is the quantity prognosed by the LES SGS closure. The log law in Figure **??** is based on a roughness length of 0.1m, a reference velocity of 10m/s, and a reference height of 500m.

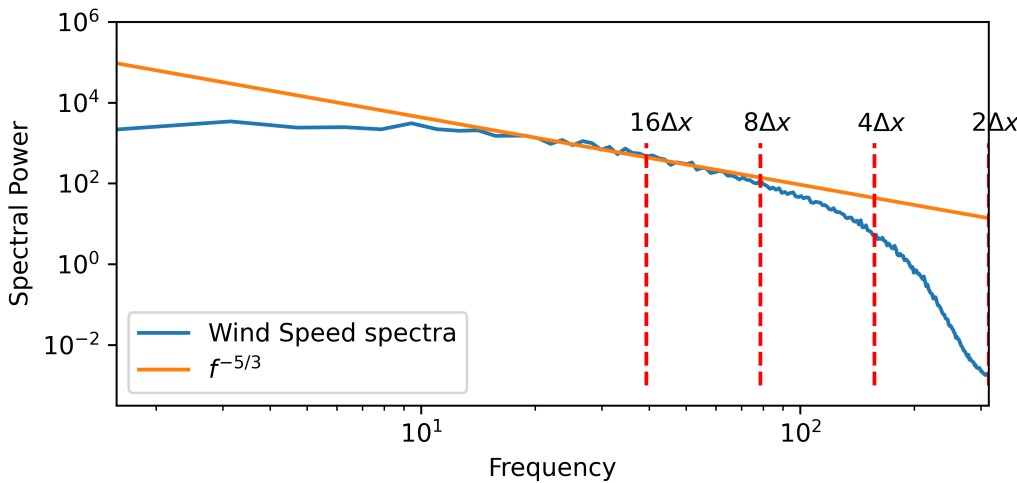

**Figure 4.** Velocity magnitude spectra at $z = 75$m and $t = 10$ hr for the neutral ABL test case, computed as the squared absolute value of transformed complex FFT values averaged over 1-D sweeps in the x and y directions.

The inversion height appears to be a bit more smoothed out and lower than WRF in Mirocha et al. (2018) and roughly the same as SOWFA and HiGrad. The inversion appears to have roughly the same smoothness as FastEddy (Sauer and Muñoz-Esparza, 2020) but is a bit lower in height. The magnitude of updrafts are similar to FastEddy in Figure 8 of Sauer and Muñoz-Esparza (2020).

Figure 6 shows the spectral power of FFTs of horizontal wind, vertical wind, and potential temperature at a height of $z = 75$m at $t = 2$hr. The resolution of potential temperature (treated quite similarly to tracers in the dynamics) is somewhat better than that of the horizontal and vertical velocities. This is likely due to differences in the diffusion where velocities use a stress tensor and tracers use a second-order diffusion operator. All of these quantities are resolved (from a spectra magnitude perspective) at a scale of roughly $4 - 8\Delta x$, keeping in mind that spectra do not always provide seemingly physically realizable results for physical reasons (Norman et al., 2023b). In this simulation, though, the flow is absent of Runge oscillations, grid point convection cells, and grid-scale energy accumulation.

Figure 7 plots mean velocity magnitude along with the mean range of one standard deviation (with regard to deviations in time), averaged between 2 and 3 hours of simulation, for the portUrb simulation along with the mean and one standard deviation range of observations from the Sandia National Laboratories Scaled Wind Farm Technology (SWiFT) field experiment (Kelley and Ennis, 2016) and four model mean velocities for comparison: WRF, SOWFA, HiGrad (Mirocha et al., 2018), and FastEddy (Sauer and Muñoz-Esparza, 2020). The portUrb results are the closest to observations, though this is partly due to a quick tuning for geostrophic wind forcing and surface heat flux values (similar to the tuning procedure of Mirocha et al. (2018) in Table 2). Two different simulations were run for portUrb: (1) with $u_g = 9$ m/s with $0.35$ K m s$^{-1}$ heat flux; and (2) $u_g = 10$ m/s with $0.4$ K m s$^{-1}$ heat flux. The latter option performed better, and those are the results shown here.

Figure 8 plots time-averaged total TKE between 2 to 3 hours as well as observed TKE and results from three other models. PortUrb gives very similar TKE results to the SOWFA code and is close to observations in $z \in [10, 50]$m. None of the LES models maintain a consistent TKE in $z \in [50, 250]$m as seen in the observations, however. This may simply be due to the observed boundary layer being more developed than the simulations' boundary layers.

## 3.3 Supercell

### 3.3.1 Specification

The splitting supercell test case (Weisman and Klemp, 1982; Droegemeier et al., 1993; Weisman and Rotunno, 2000; Morrison and Milbrandt, 2011; Klemp et al., 2015; Zarzycki et al., 2019) simulates a vertical profile with significant Convectively Available Potential Energy (CAPE), vertical shear, and veering with convection initiated by a positive temperature perturbation. The horizontal velocities as well as water vapor dry mixing ratio are initialized according to the sounding used by Morrison and Milbrandt (2011) that is specified in the WRF 4.6.1 idealized test case titled "quarter_ss". The horizontal velocities follow a "quarter circle" hodograph up to 2.3km with a continued linear shear in the $x$-direction up to 7km. The total sheer length is 40km. Different models commonly differ substantially in solutions as seen in the supercell inter-comparison between different microphysics choices (Morrison and Milbrandt, 2011) and supercell inter-comparison between dynamical cores with identical microphysics (Zarzycki et al., 2019) on a reduced radius sphere. One important difference between this study and that of Morrison and Milbrandt (2011) is that the time step size for microphysics in this study is about $10\times$ smaller, meaning microphysics is called about $10\times$ more frequently. The microphysics time step size is known to have a strong influence on the simulation. This study also uses much higher-order-accurate reconstructions in the Finite-Volume approach than what is used in Morrison and Milbrandt (2011). Also, it is unclear if the study in Morrison and Milbrandt (2011) uses a sub-grid-scale turbulence closure similar to the one used in this study.

The initial potential temperature profile of Klemp et al. (2015) used in the WRF idealized supercell test case essentially sets up a constant lapse rate from 300K at the surface to 213K at the tropopause. Since this model's coupler state is based on temperature, that constant lapse rate is used directly for simplicity:

$$T_{initial}(z) = \begin{cases} T_0 + \left( \frac{T_{trop} - T_0}{z_{trop}} \right) z & \text{if } z \leq z_{trop} \\ T_{trop} & \text{if } z > z_{trop} \end{cases} \tag{21}$$

where $z_{trop} = 12$km, $T_0 = 300$K, and $T_{trop} = 213$K. Water vapor dry mixing ratio ($q_v$), $u$-velocity, and $v$-velocity are all interpolated from a table provided by WRF 4.6.1, specifying the hodograph and moisture used in Morrison and Milbrandt (2011). With temperature and water vapor dry mixing ratio specified, the hydrostatic relationship and the moist equation of state are combined to form a hydrostatic integration of the full pressure as follows:

$$\frac{d\ln p}{dz} = -\frac{(1 + q_v) g}{(R_d + q_v R_v) T} \implies p_2 = p_1 \exp\left( -\int_{\zeta=z_1}^{\zeta=z_2} \frac{(1 + q_v(\zeta)) g}{(R_d + q_v(\zeta) R_v) T(\zeta)} d\zeta \right) \tag{22}$$

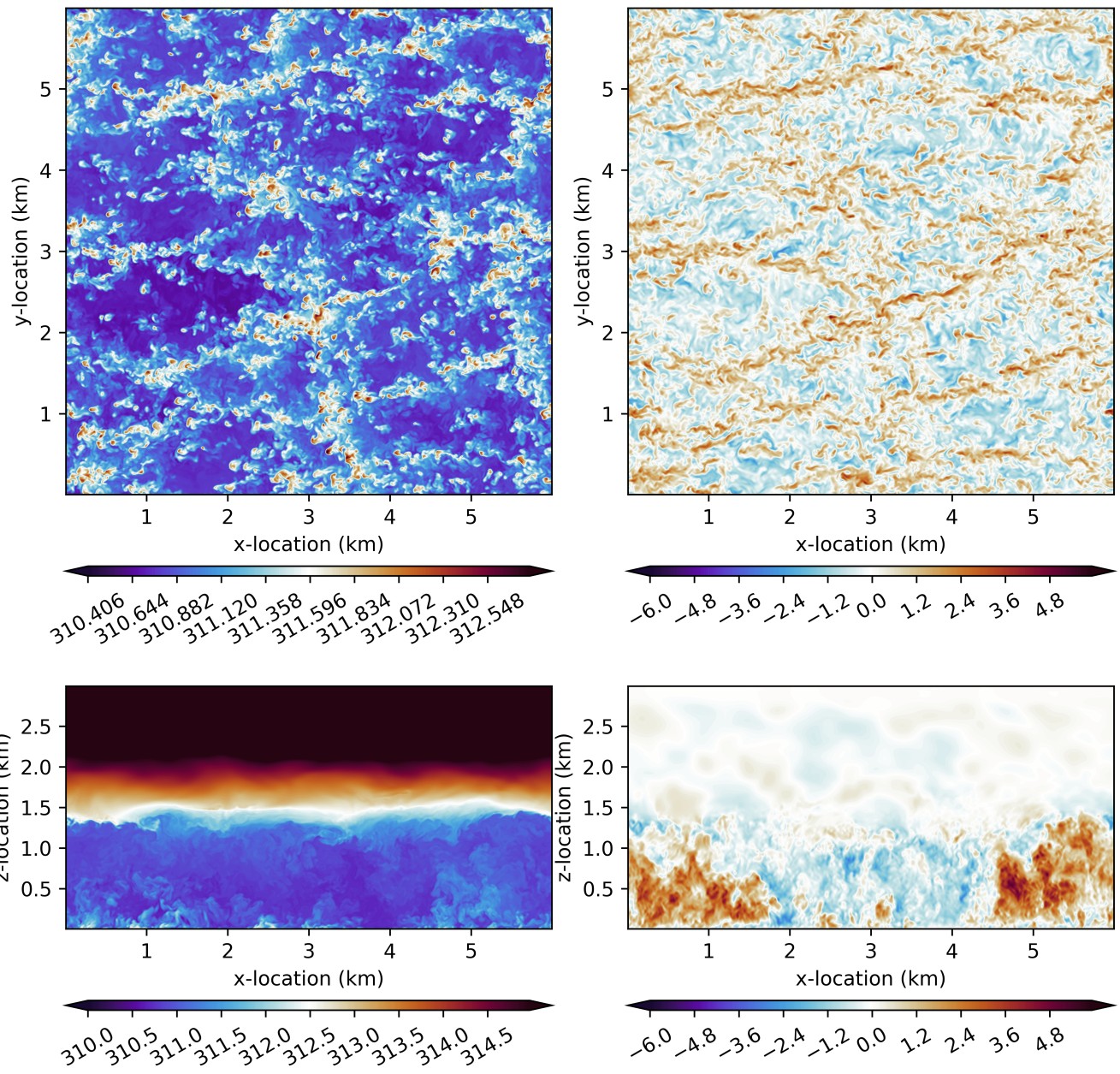

**Figure 5.** For the convective ABL test case, plots of potential temperature (K) at $z = 75$m (top left); vertical velocity (m/s) at $z = 75$m (top right); potential temperature (K) at $y = 3$km (bottom left); and vertical velocity (m/s) at $y = 3$km (bottom right) at $t = 2$hr.

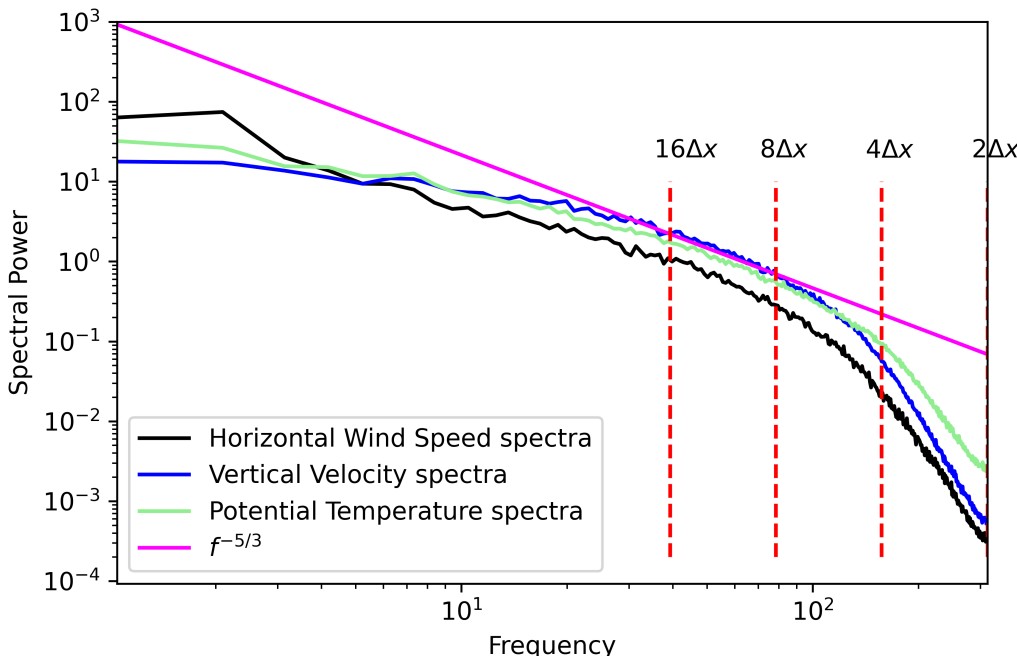

**Figure 6.** For the convective ABL test case, plot of spectral power as the squared absolute value of transformed complex FFT values as an average of 1-D sweeps in the $x$ and $y$ directions for various quantities at $z = 75$m at $t = 2$ hr.

Initialization uses 9-point GLL quadrature within each cell $\Omega_k$, and 9-point GLL quadrature is used to integrate the above hydrostatic relationship within the intervals created by the first set of nine GLL points within cells. Surface pressure is assumed to be $p_0 = 10^5$ Pa. This results in a hydrostatically balanced initial state. With pressure, temperature, and water vapor dry mixing ratio specified at GLL points within each cell, the dry air and vapor densities can be recovered, providing initialization of all model variables at GLL points within each cell.

A domain of $200 \times 200 \times 20$ km is used to simulate for two model hours at a grid spacing of 500m in the vertical direction and 1km in the horizontal directions, running the microphysics every dynamical core time step of 0.267 seconds (limited by an acoustic CFL value of 0.6). The portable C++ microphysics were ported from the Fortran Morrison 2-moment microphysics of WRF 4.6.1 with default settings. Run side-by-side in a supercell simulation, the accelerated physics on an Nvidia A100 GPU give a mean of $10^{-9}$ relative differences per time step or less for all moist species, number concentrations, and temperature updates. All moist species and number concentrations are transported by the dynamical core. All mass-providing moist species were also added to the total density in the dynamical core for proper mass-loading in buoyant and advective dynamics. The "ihail" option is enabled in the code, meaning hail-like graupel parameters are used.

The top 2% of the domain is a sponge layer nudging temperature and velocities toward initial values with a decay of $z^3$ away from the top boundary. The top and bottom boundaries are free-slip solid walls with an isentropic thermal condition, and

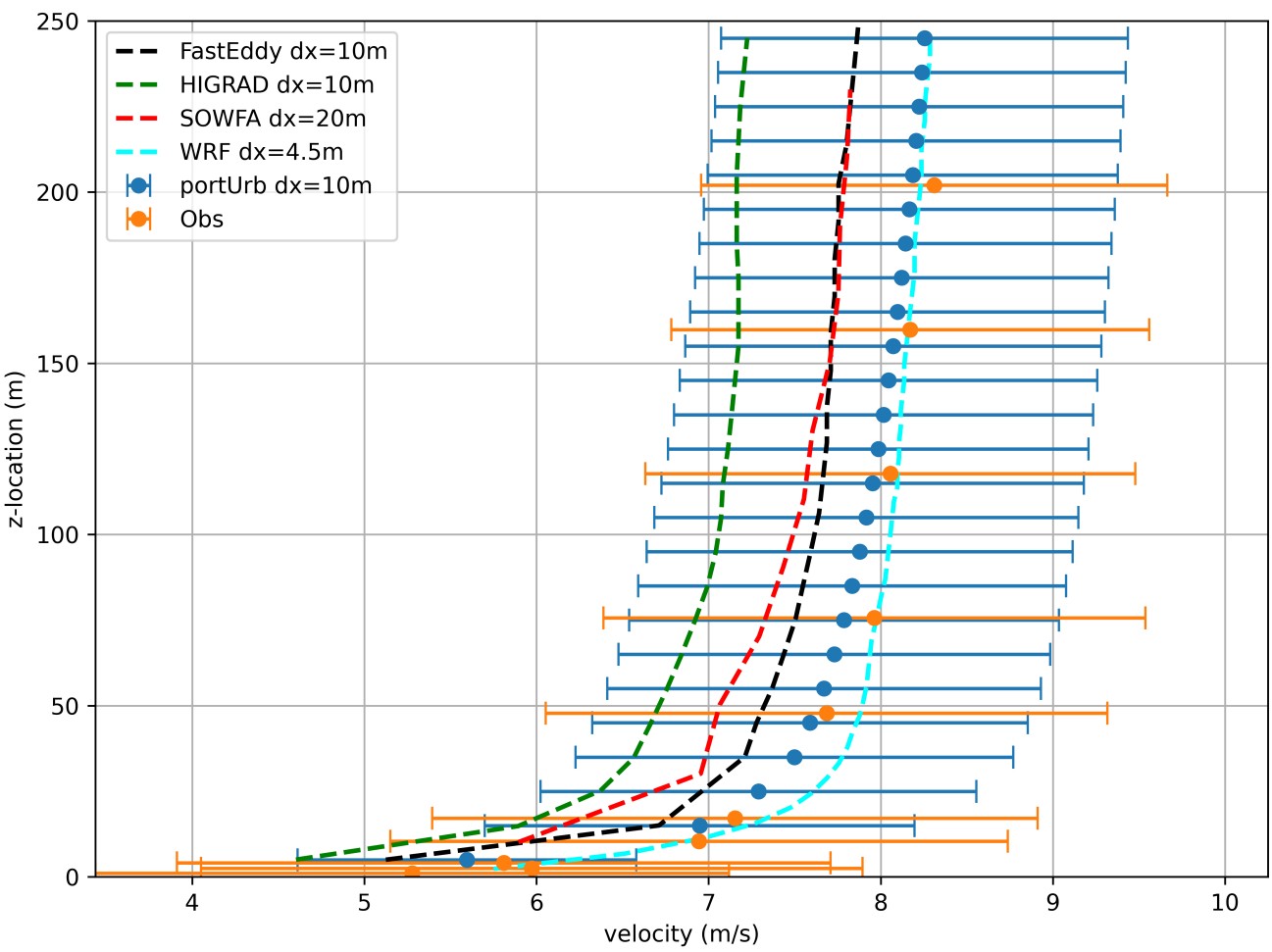

**Figure 7.** For the convective ABL test case, 2-3hr time-averaged velocity magnitude (m/s) and the mean range of one standard deviation (in time) for portUrb (the model from this study) and observations (Kelley and Ennis, 2016); as well as four other model means (Sauer and Muñoz-Esparza, 2020; Mirocha et al., 2018) as references.

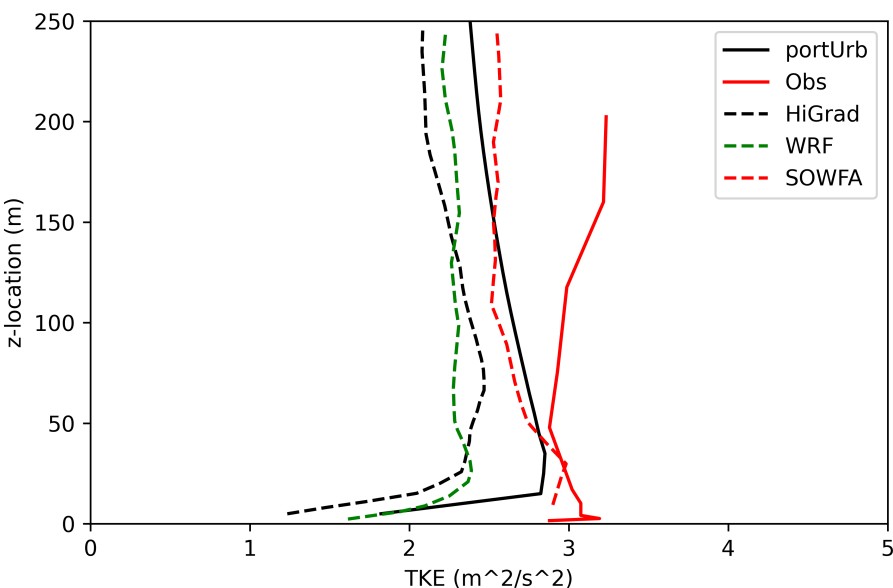

**Figure 8.** For the convective ABL test case, 2-3hr time-averaged total TKE for portUrb (the model from this study) and observations (Kelley and Ennis, 2016); as well as three other model means (Mirocha et al., 2018) as references.

the horizontal boundaries are periodic. No surface fluxes are applied in this simulation. Output is performed every 60 model seconds.

### 3.3.2 Results

Figure 9 plots iso-surfaces of different cloud and precipitation properties at 30, 60, 90, and 120 minutes of simulation. The initial warm thermal rises and begins to precipitate. As the storm evolves (asymmetrically due to the quarter circle hodograph), the initial thermal splits into two cells in the $y$-direction.

Figure 10 plots time traces of minimum surface potential temperature perturbation, $\theta'$, defined as departure from the initial state; the portion of the surface inside the cold pool, defined as $\theta' < 2K$; the accumulated surface precipitation; the maximum updraft velocity; and the minimum low-level downdraft ($z < 3.5$km). The minimum $\theta'$ is very similar to Figure 6a of Morrison and Milbrandt (2011) in terms of the trend and magnitude, and it is within the variability at 0, 60, and 120 minutes of simulation time plotted in light blue in Figure 10. While surface cold pool fraction is a bit lower than Figure 6b in Morrison and Milbrandt (2011), it is in line with the variability of that study shown in light blue shading. The surface accumulated precipitation is also in line with the variability in Morrison and Milbrandt (2011). The minimum low-level downdraft and maximum global updraft over time is quite similar in both trend and magnitude to the MOR-BASE configuration of Morrison and Milbrandt (2011) shown using dashed lines in Figure 10d.

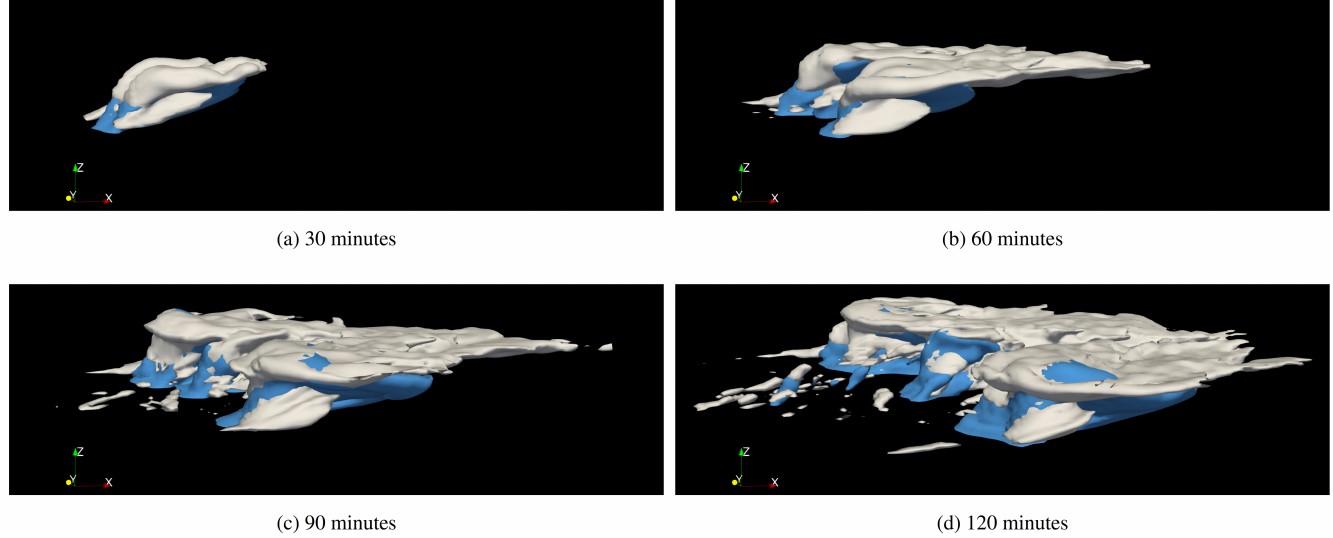

(a) 30 minutes                      (b) 60 minutes

(c) 90 minutes                      (d) 120 minutes

**Figure 9.** For the supercell test case, iso-surfaces of the sum of cloud liquid, cloud ice, and cloud snow densities (white) at a value of $0.0005 \text{ kg m}^{-3}$; and the sum of rain water and graupel densities (blue) at a value of $0.001 \text{ kg m}^{-3}$.

Figure 11 shows horizontally averaged microphysics quantities after one and two hours of simulation along with reference plots of the MOR-BASE configuration of Morrison and Milbrandt (2011) in dashed lines. The rain values, $q_r$, are quite similar to those of MOR-BASE. Cloud liquid, $q_c$, also has very similar structure and magnitude. In these simulations, the height to which cloud ice and precipitated species are forced is lower than shown in Morrison and Milbrandt (2011). Given the similar updraft speeds, this could be due to less diffusion of the temperature inversion in this study. Figure **??** suggests at the second temperature discontinuity that the diffusion of features like this is low. In this study, the inversion height grows by quite a bit between one and two hours of simulation, while in Morrison and Milbrandt (2011), the inversion height doesn't appear to change significantly. This study has lower amounts of frozen species aloft, but the height of extrema in this study follow the same structure as those in Morrison and Milbrandt (2011) shown using dashed lines in Figure 11.

### 3.4 Staggered Surface-Mounted Cube Array

This test case compares LES solutions of flow past a large array of surface-mounted cubes against results from a wind tunnel experiment (Castro et al., 2006; Tomas et al., 2016; Muñoz-Esparza et al., 2020). The simulation approximates conditions at the center of the cube array, where measurements were made, by using periodic horizontal boundary conditions with four staggered cubes. The cubes are of a length, width, and height of $h = 20$mm. The domain is $4h \times 4h \times 10h$ in the $x$, $y$, and $z$ directions, respectively. The bottom and top boundaries are free-slip solid walls with isentropic thermal conditions. A kinematic viscosity of $\nu = 1.5 \times 10^{-5} \text{m}^2 \text{s}^{-1}$ is used, and a grid spacing of 1mm is used in all directions. Output is performed every 0.1s, and the simulation is run for 5s total.

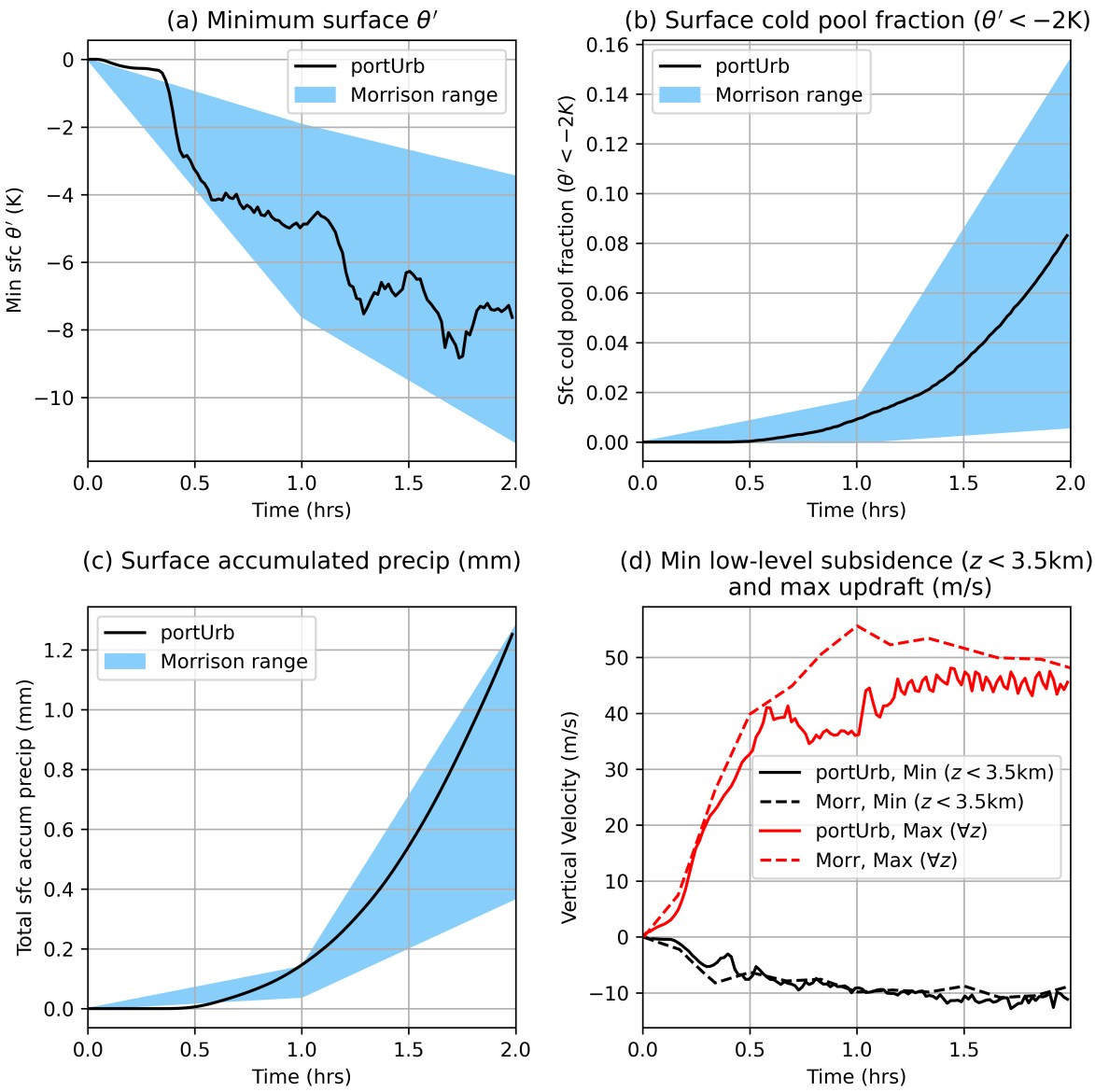

**Figure 10.** For the supercell test case, traces of various quantities over the course of the two-hour simulation. In sub-figures a, b, and c, the shaded region denotes the range over perturbations of different microphysics options in Morrison and Milbrandt (2011). In sub-figure d, "Morr" represents the MOR-BASE data from Morrison and Milbrandt (2011) since values were not given for each perturbation of microphysics options.

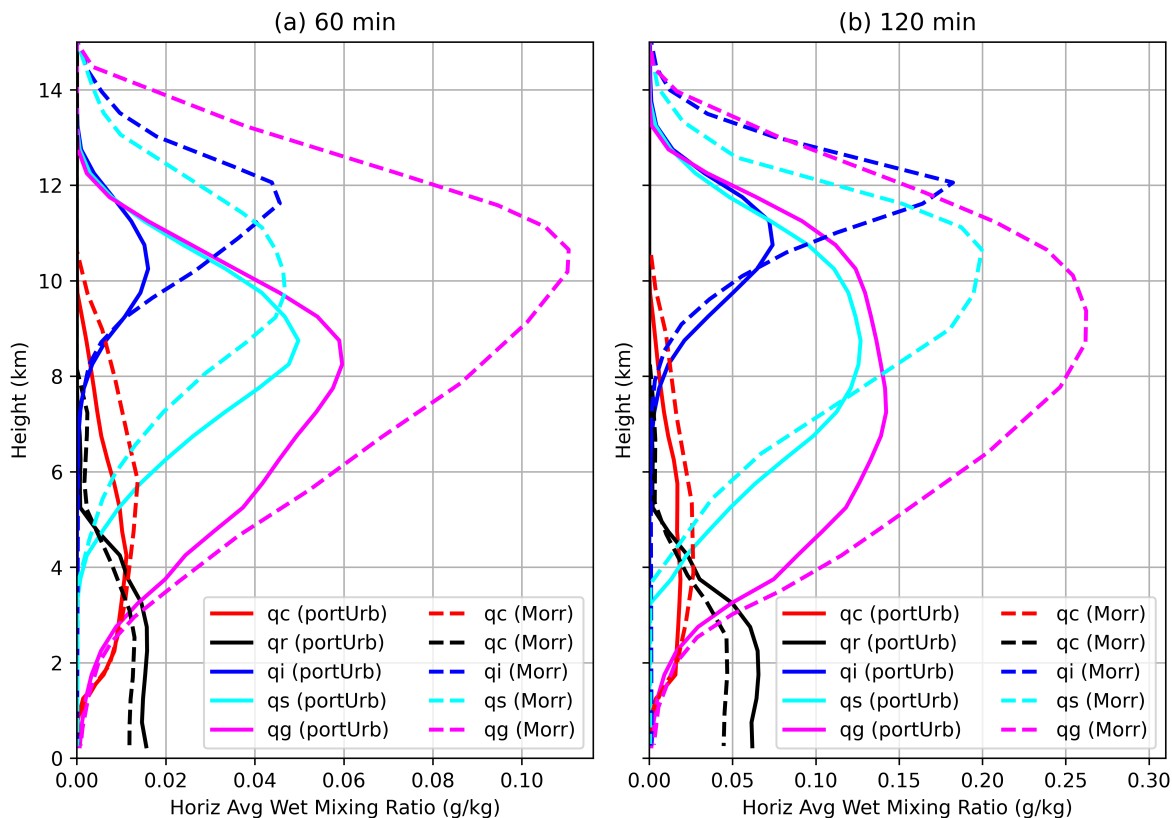

**Figure 11.** For the supercell test case, horizontally averaged cloud and hydrometeor wet mixing ratios at one and two model hours. $qc$ is cloud liquid, $qr$ is precipitated rain, $qi$ is cloud ice, $qs$ is precipitated snow, and $qg$ is precipitated graupel. "Morr" comparison points denotes the values for the MOR-BASE simulations in Morrison and Milbrandt (2011).

Gravity is not enabled for this simulation, initial pressure is $10^5$ Pa, initial temperature / potential temperature (they are the same in this case) is 300K, and initial velocity is 10m/s in the $x$ direction and zero in the $y$ and $z$ directions. A pressure gradient forcing is used to penalize deviations from 5.82 m/s in the $x$ direction at a height of 0.043m (the average of the observed velocities at the highest points above the ground) over a time scale of 100 time steps. A surface flux is applied with a roughness length of $10^{-7}$ m at the model surface and $10^{-6}$ m at cube surfaces. Surface roughness lengths in the set $\left\{10^{-7}, 10^{-6}, 10^{-5}\right\}$

were tested, fully tensored in 9 experiments, for the model surface and cube surfaces using portUrb's ensemble capability.

The four cubes are placed on the following domains: (1) $[h/2, 3h/2] \times [h/2, 3h/2] \times [0, h]$; (2) $[h/2, 3h/2] \times [5h/2, 7h/2] \times [0, h]$; (3) $[5h/2, 7h/2] \times [3h/2, 5h/2] \times [0, h]$; and (4) $[5h/2, 7h/2] \times [0, h/2] \times [0, h] \cup [5h/2, 7h/2] \times [7h/2, 4h] \times [0, h]$. In the experiment, the flow is sampled at various vertical columns at four horizontal locations, each of which have four corresponding points of geometric similarity in the domain. The four point locations are: (1) $P_0 = (h, 3h)$, which corresponds to the center of

a cube location; (2) $P_1 = (2h, 3h)$; (3) $P_2 = (h, 2h)$; and (4) $P_3 = (2h, 2h)$.

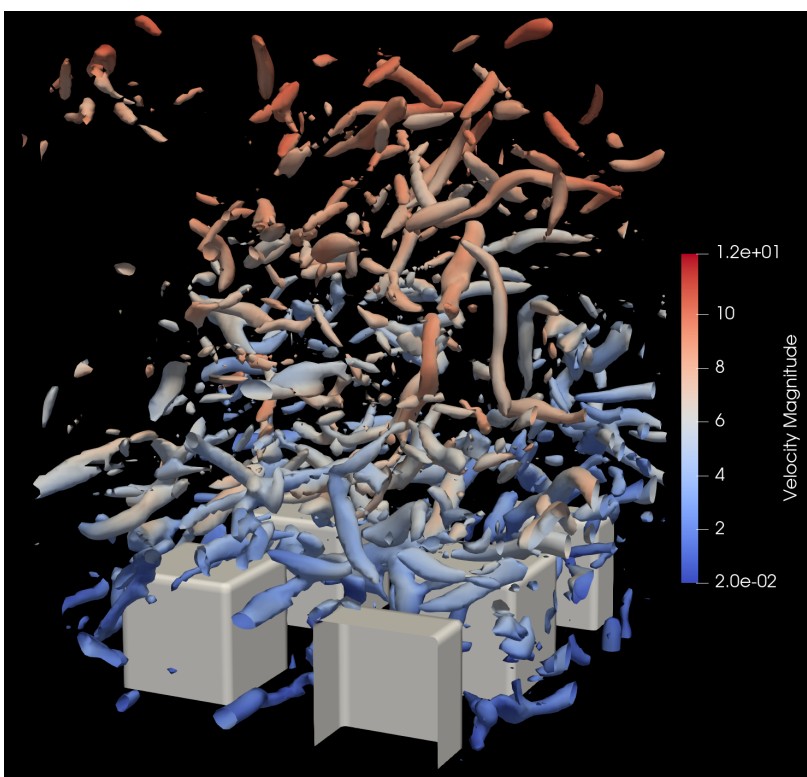

**Figure 12.** For the staggered surface-mounted cube array, iso-surfaces of the cube surfaces (white), and iso-surfaces of Q-criterion at a value of $3 \times 10^{-5} \ \text{s}^{-2}$ with color shading coming from wind speed (m/s).

Given there are four cubes, each of these points have three other points that are geometrically identical inside the domain (due to periodic horizontal boundary conditions and symmetry of cube placement within the domain), creating four points total at which $P_0$, $P_1$, $P_2$, and $P_3$ will be each be sampled – the results of which are averaged in order to compare against wind tunnel observations. These other three geometrically identical points correspond to shifts of $(-2h, 0)$; $(2h, h)$; and $(2h, -h)$ – where periodic wrapping is used to remain in the domain. Where points lie on a domain boundary, the smallest location magnitude (zero) is used.

The time-averaged $u$-velocity and $\overline{u'w'}$ outputs over intervals of 0.1s are all interpolated to observation locations and compared against observations to determine the best fits. The one second period with the best fit according to a relative $L_2$ norm averaged among the six comparisons (four $\overline{u}$ comparisons and two $\overline{u'w'}$ comparisons) is used to compared against observations. The time period $[0.4\text{s}, 1.4\text{s}]$ is used for comparison.

Figure 12 shows the placement of cube isosurfaces as well as Q-criterion to visualize the vortices in the flow at $t = 1$ second. Figure 13 plots these quantities against observations at each of the points. The perturbation quantities $u'$ and $w'$ are temporal deviations from the current simulation-spanning temporal mean and are accrued at each grid cell and each time step. The time-averaged $u$ velocity and $\overline{u'w'}$ correlations are both well captured by the LES simulation during the performance time

period. The $\overline{u'w'}$ extrema above the cube surface at $P_1$ is not well-captured in terms of scale (an error common to other LES simulations of this case as well), but the structure and existence of a local maximum is captured. The correlation at $P_3$ is very well captured up to 1.5 cube heights (particularly the slope up to the maximum and the maximum magnitude as well), but it is underestimated above that height. In summary, comparison against anemometry data shows that the LES model, the immersed boundary implementation, and the surface friction implementation are well formulated for capturing realistic

turbulent dynamics around obstacles.

### 3.5    Coarsely Resolved Sphere

In order to tune the immersed boundary approach for partially immersed cells, a sphere is placed in a domain with periodic conditions in the $y$ and $z$ directions with specified laminar inflow / outflow boundaries in the $x$ direction and the gravity term (and hence thermal and mass stratification) turned off. The domain is 2km $\times$ 1km $\times$ 1km in the $x$, $y$, and $z$ directions,

respectively. The sphere center is located at 0.4km in the $x$ direction and is centered in the $y$ and $z$ directions with a radius of 100m. A wind velocity of $u = 10$ m/s, $v = w = 0$ is specified at $x$-direction boundaries through a horizontal sponge layer over 2% of the domain length with a decay of $x^3$ to serve as inflow and outflow non-reflecting boundaries.

     A high resolution reference solution is performed at 10m grid spacing (20 cells across the sphere diameter), and low resolution runs are performed at 50m grid spacing (4 cells across the sphere diameter) to test different exponents to the immersed

proportion in equation (16). Testing integer increments of the exponent in equation (16), the value of five was demonstrated to give the lowest errors relative to the high resolution time-averaged solution coarsened to the low resolution grid. The time average is taken between 15 and 60 minutes of a 60 minute simulation. The sum of differences for each individual time-averaged velocity component is used for the comparison between high and resolution solutions.

     A plot of stream tubes flowing past the sphere at high and low resolution is given in Figure 14. The low resolution solution has

lower velocities because the cells near the sphere are coarser and have partially immersed material in them, slowing down the flow near the sphere. The flow slightly further away from the sphere is also averaging in lower wind speeds in the coarser-scale cells.

### 3.6    Flow Through Manhattan Buildings

To observe the character of the flow through realistic buildings and the influences of building friction, this test case performs

flow through a subset of buildings in Manhattan southwest of Central Park, rotated by 29° (to achieve a roughly North-South orientation of the buildings) over a domain of $1.31 \times 1.716 \times 0.572$ km in the $x$, $y$, and $z$ dimensions. There are two primary goals of these simulations: (1) to understand if the code is well-posed for this kind of flow, free from significant numerical artifacts, and giving physically realizable results; and (2) to demonstrate the influences of different building roughness lengths on flow through and above the buildings. Eleveth-order WENO interpolation is used for this simulations with no modifications

from the other simulations in this study.

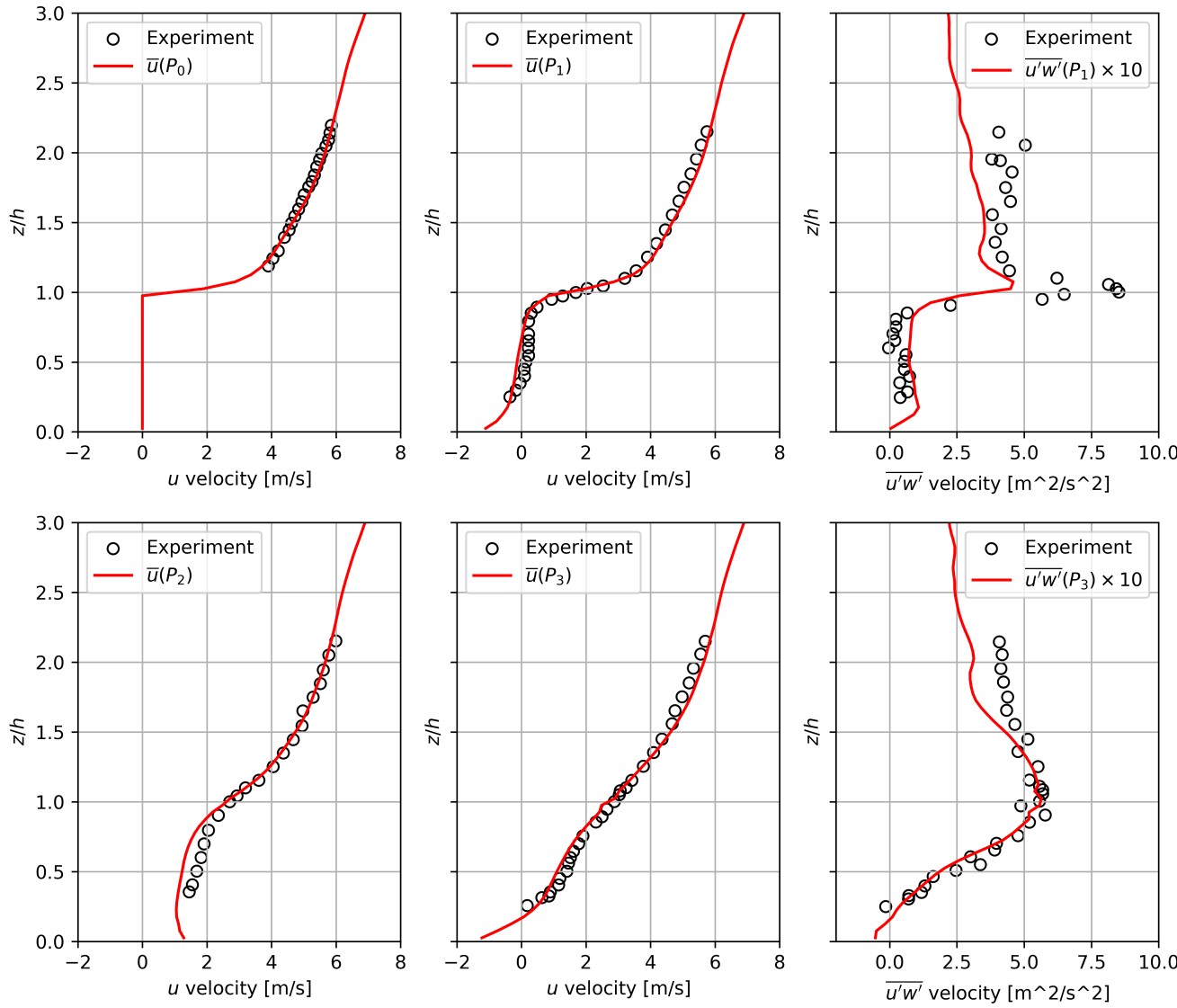

**Figure 13.** For the staggered surface-mounted cube array test case, plots of time-averaged $\overline{u}$ over the time domain $[0.4\mathrm{s}, 1.4\mathrm{s}]$ at all four points, time-averaged $\overline{u'w'}$ at points $P_1$ and $P_3$, and experimental results. Perturbations $u'$ and $w'$ are deviations from the current temporal average at a given instance in the simulation.

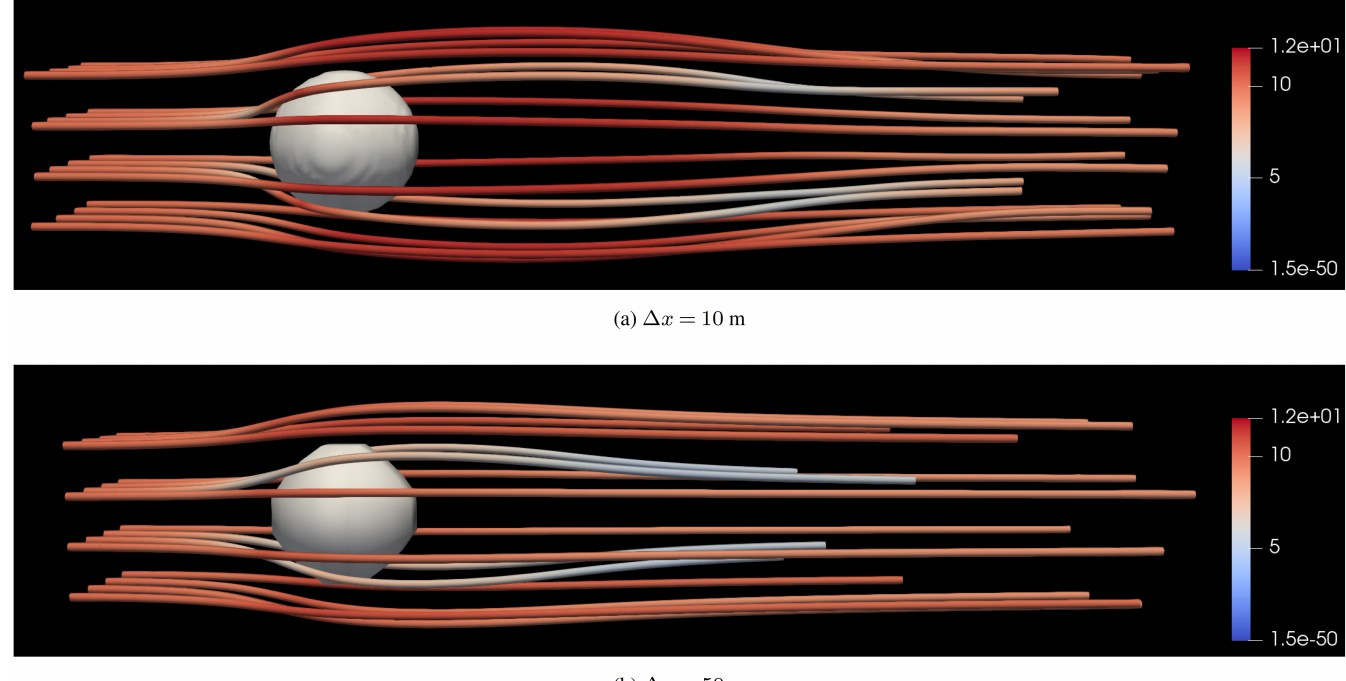

(a) $\Delta x = 10$ m

(b) $\Delta x = 50$ m

**Figure 14.** For the flow past a sphere test case, stream tubes are plotted in a 4x4 grid of points flowing past the sphere at high resolution and at low resolution, colored by the velocity magnitude. The sphere is represented with isosurfaces in white at a threshold of 0.5 immersed proportion.

To ingest building geometry, the Blosm[2] plugin to the application Blender[3] is used to bring in 3-D building geometry from OpenStreetMap[4][5] (the data from which is available under the Open Database License or ODbL), which automatically transforms WGS84 latitude-longitude coordinates into local East-North-Up (ENU) Cartesian coordinates in units of meters. Buildings outside the rectangular horizontal area of interest are removed, and the file is saved as a triangulated Wavefront "obj" file (meaning only triangles are allowed as surfaces rather than generic polygons). The triangulated mesh is then read into portable C++ using custom code, ignoring material properties and vertex normals – essentially only storing a set of triangular faces, each defined by three vertices in 3-D space.

Finally, for each horizontal point over a $9 \times 9$ point Gauss-Legendre-Lobatto (GLL) grid of points within each horizontal cell, the maximum height of any intersecting mesh triangle at a given point is computed and stored. During initialization, anything below that maximum point is declared to be embedded, and the proportion of embedded material in each cell is computed with GLL quadrature in all dimensions. While this process could potentially be improved by doing raycasting to determine non-

---

[2]https://github.com/vvoovv/blosm
[3]https://www.blender.org/
[4]https://www.openstreetmap.org/
[5]https://openstreetmap.org/copyright

monotonically embedded regions in the vertical, this requires a well-posed mesh to avoid incorrect detection of which portions of the vertical are immersed and which portions are not. Because of the extra mesh requirements and reliance on external tools to create these meshes, this is left for future research. The extension is straightforward, however, as raycasting is a relatively cheap and simple process.

A 20m padding is added to the horizontal direction – roughly the mean distance between blocks of buildings in this domain, and a 100m padding is added above the highest building in the domain, which is 470m. A neutrally buoyant atmosphere is specified throughout the vertical domain with a sponge layer applied to the top 2% of the domain to absorb waves that would otherwise reflect. A wind speed of 20 m/s from the West (rotated by $29°$ to account for rotated building geometry) is initialized everywhere except in immersed cells. Deviations from a horizontal mean of 20 m/s flow from the West at a height of 500m are penalized to enforce a consistent wind speed throughout the course of the simulation (i.e., pressure gradient forcing). Periodic boundary conditions are applied in the horizontal direction to quickly spin up turbulence, essentially assuming infinite repetition of the city block geometrically. In the vertical direction, free-slip boundaries are applied with a surface friction at the bottom boundary specified by a roughness length of 0.05m.

Three different building material roughness lengths are used (all sharing the same surface roughness length): $10^{-6}$m, 0.05m, and 0.5m. The value of $10^{-6}$m is essentially free-slip when using a grid spacing of 2m; 0.05m is likely the roughly physical value (accounting for unresolved building features like window insets, bricks, etc.); and 0.5m is likely unrealistically large. Simulations are performed for 100 minutes: a 10-minute spin-up period followed by a 90-minute simulation period for time-averaging and comparison between different building roughness lengths. On 120 Nvidia A100 GPUs, the 100-minute simulations took about 24 hours of walltime a piece to complete, with about 1.3 million cells per GPU (161 million cells total). 2.25 million time steps were required to simulate 100 minutes in model time.

Figure 15 gives iso-surface plots of the immersed proportion (i.e., buildings) in white, and Q-criterion in blue (shaded by velocity magnitude) at $t$=100 minutes of simulation using a building surface roughness of 0.05m. Because Q criterion is plotted at a positive iso-surface value, vortex-dominated flow is shown rather than shear-dominated flow – thus the gap between buildings and visualized vortices as sheared flow transitions to turbulent vortices on the lee side of buildings.

Figure 16 plots, at $z = 1$m, the 90-minute time-averaged velocity magnitude at a building surface roughness of $10^{-6}$m, ($\left|\boldsymbol{u}\left(z_0 = 10^{-6}\right)\right|$), where discontinuities are the most likely to show up in the velocity time average. It also plots the difference, $\left|\boldsymbol{u}\left(z_0 = 10^{-6}\right)\right| - \left|\boldsymbol{u}\left(z_0 = 0.05\right)\right|$, as well as the difference, $\left|\boldsymbol{u}\left(z_0 = 10^{-6}\right)\right| - \left|\boldsymbol{u}\left(z_0 = 0.5\right)\right|$, to show how much slower the flow through frictioned buildings is. This is also plotted for $z = 101$m in Figure 17. In these plots, there is no sense of any discontinuities present in the solution at either of the vertical levels. As the building friction increases, as expected, the flow becomes slower through the buildings, though not universally so (even in the 90-minute time average) due to the turbulent nature of the flow. Also, as expected, the decrease in flow velocity is concentrated more highly at building edges and is more diffuse farther away from buildings.

Street canyon ventilation is generally greater with less frictioned buildings as expected, meaning pollutants and temperature extrema are likely more effectively mixed out of the surface layer. This is evidenced by the velocity magnitudes being larger with lower friction but also by the increased resolved and unresolved TKE shown in Figures 18 and 19, respectively. These plots

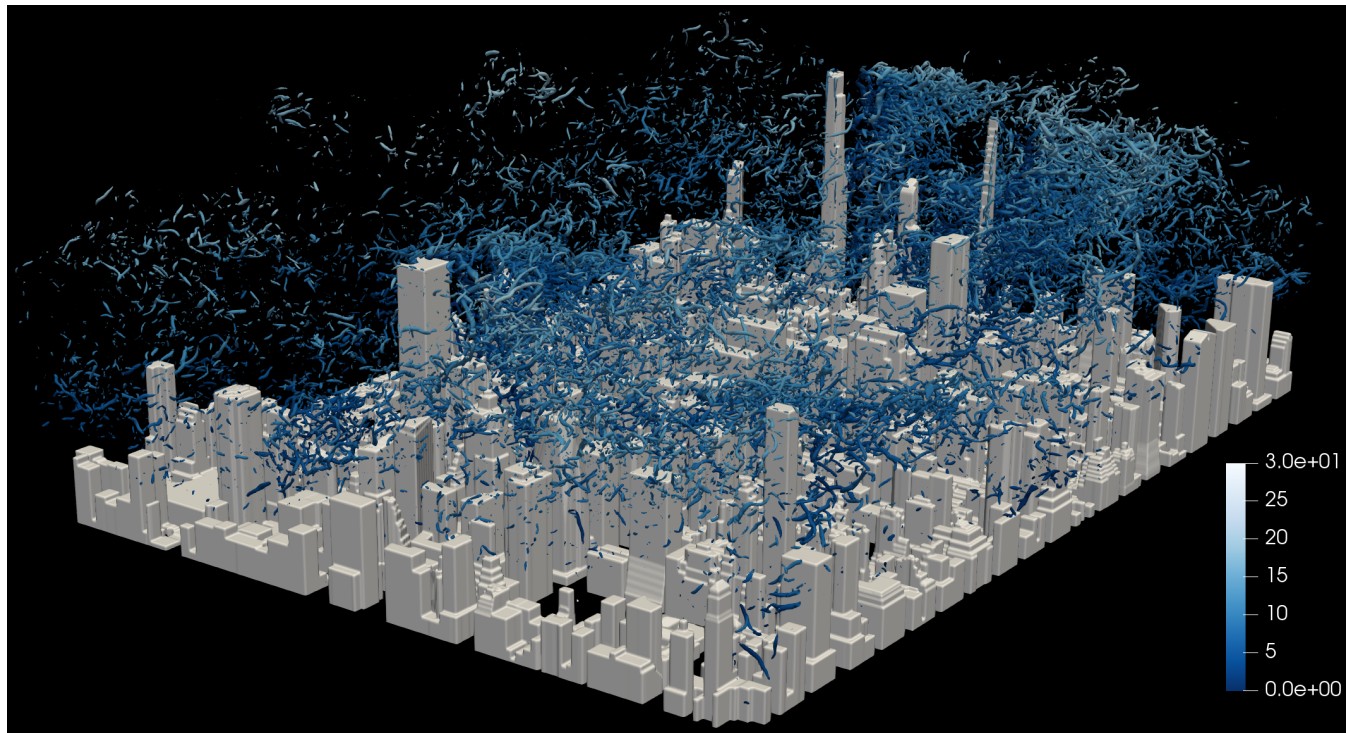

**Figure 15.** Iso-surface plots of flow through Manhattan buildings with 2m grid spacing and a building surface roughness of $z_0 = 0.05$m after 100 minutes of simulation. Immersed proportion iso-surfaces are given in white. Q-criterion iso-surfaces are at $1\ \text{s}^{-2}$, colored by velocity magnitude (m/s).

indicate that fluctuations are also generally greater with lower building surface friction. There is a location at around $x \approx 260$m, $y \in [500, 600]$ m, where the surface experiences significant mean velocity and fluctuations in the 90-minute average, showing an interesting convergence from induced vertical flow through the buildings. It is not the only location of this nature, but it is the most significant, showing that mean surface-level flow can be quite intermittent spatially in urban environments both in mean values and in fluctuations.

While a realistic flow through urban settings would clearly require tuning the building and surface friction terms to match observed flow, this test case demonstrates physically realizable results coupled with expected outcomes regarding mean flow rate and fluctuations in an urban setting when perturbing the building surface friction settings with realistic geometry. Also, it appears that complex building geometry can be ingested directly into the model without any need for smoothing, pre-processing, or conditioning. The use of WENO limiting, an upwind Riemann solver, and slight softening of building edge immersed cells in terms of forcing time scale appear to be enough to handle any generic building geometry while avoiding visible artifacts in the solution.

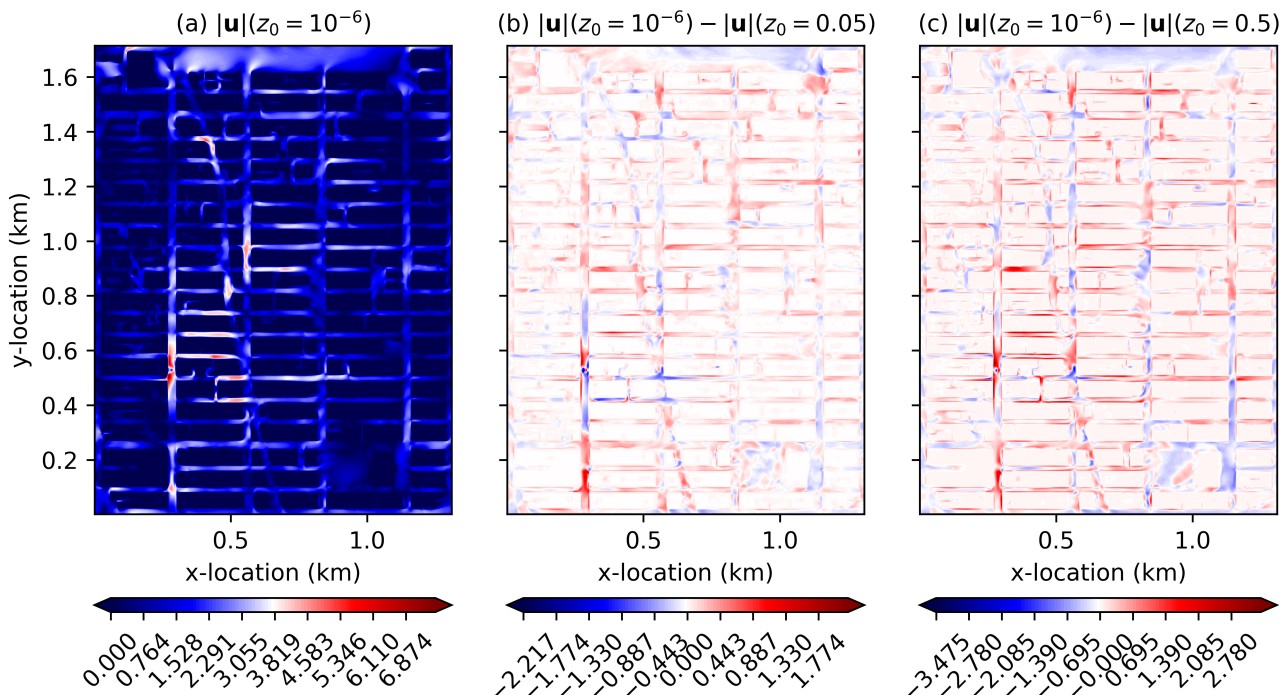

**Figure 16.** For the flow through Manhattan buildings test case, plots of 90-minute time-averaged wind speed magnitude and differences (m/s) for different building roughness lengths at a height of $z = 1$m (the surface layer of cells) for flow through Manhattan buildings.

### 3.7 City Flow Forced by Turbulent Precursor

This test case simulates the same building setup as section 3.6 except with a larger domain and the use of a concurrent turbulent precursor for forcing the inflow boundaries, open boundaries for outflow, and a stretched vertical grid reaching up to 1,800m. The city setup covers a domain of $1{,}269 \times 1{,}675$m $\times 472$m in the $x$, $y$, and $z$ dimensions, respectively. To allow for a well-posed turbulent precursor forcing, the domain size is doubled in the horizontal dimensions with the city placed in the center; and the vertical domain is extended to 1,800m in order to simulate the interaction of the city with the ABL inversion layer.

The horizontal grid spacing is set to 4m. The vertical grid is stretched from approximately 4m grid spacing in the domain $z \in [0, 480]$m to a grid spacing of 10m in the domain $z \in [600, 1800]$m, where $z \in [480, 600]$m is the transition region. The transition region is over approximately 18 cells, and the grid spacing transition region is plotted in Figure 20. The same initial conditions in the neutral ABL test case in section 3.1 is used for this simulation as well including the geostrophic forcing to create an Ekman spiral. The simulation uses 137 million cells simulated over ten model hours to fully develop the ABL.

Figure 21 plots the instantaneous horizontal wind speed at $t = 10$hr for the concurrent turbulent precursor simulation (left) and the forced simulation with the city included (right) at $z = 100$m (top) and at $y = 1{,}676$m (bottom). From these plots, it is clear that the precursor forcing is working correctly, as the same wind patterns are observed in the inflow at the left of the top and bottom plots. The open boundary conditions in the forced city simulation are also allowing the wind to flow outward

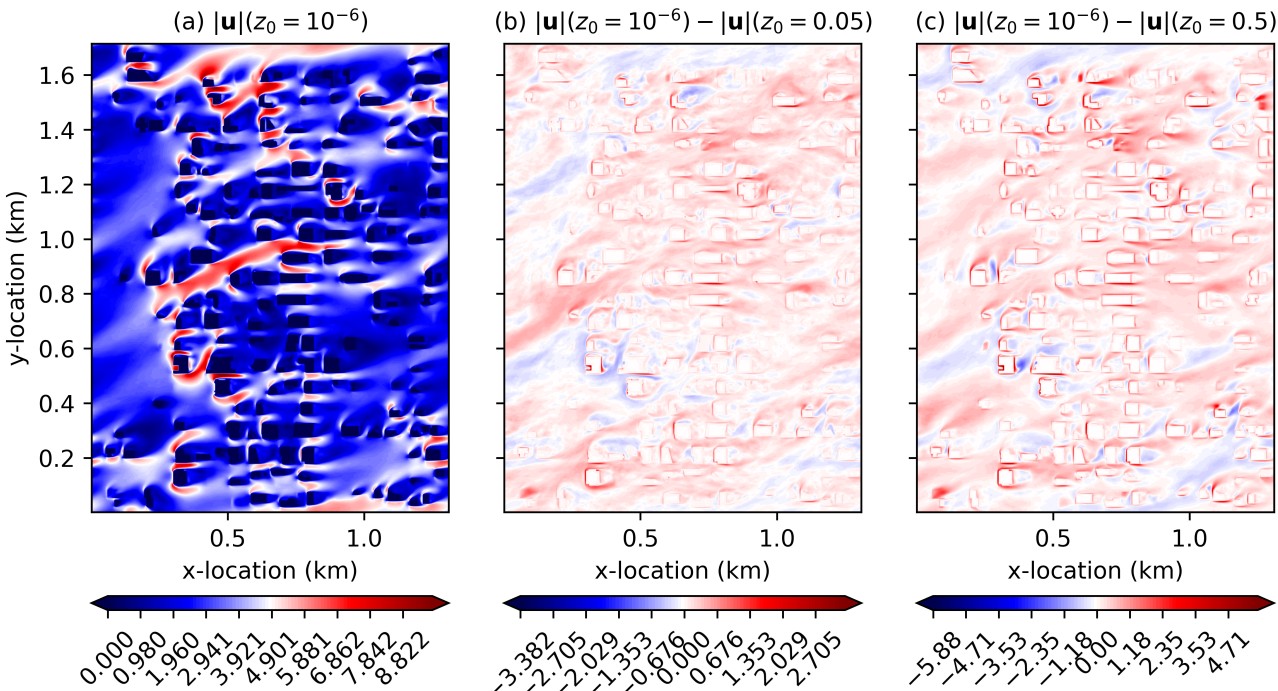

**Figure 17.** For the flow through Manhattan buildings test case, plots of 90-minute time-averaged wind speed magnitude and differences (m/s) for different building roughness lengths at a height of $z = 101$m for flow through Manhattan buildings.

without artifacts. As expected, the strongly stable ABL inversion remains intact in the precursor simulation, while it is mixed in the forced city simulation due to mechanically generated turbulence from the tall buildings, leading to gravity wave propagation in the stable layer above the inversion. Also, there is no evidence of artifacts arising from the stretched vertical grid in either the precursor or the forced simulations.

Figure 22 plots instantaneous potential temperature at $t = 10$hr for the concurrent turbulent precursor simulation (left) and the forced simulation with the city included (right) at $y = 1,676$m. From here, again, it is clear that the stable inversion remains intact in the precursor simulation and is mixed by turbulence generated by the buildings in the forced city simulation. Further, warmer potential temperature mixed downward from the stable boundary layer leads to slight warming in the leeward side of buildings where the vertical velocity is prone to descent.

Figure 23 plots the time-average over $t \in [8, 10]$hr of $u$-velocity (left), $v$-velocity (middle), and potential temperature (right) averaged over the city domain in the horizontal for precursor simulation and the city simulation forced by the precursor. From here, it is clear that the presence of the city leads to a reduction in wind speed in the atmospheric boundary layer as well as an increase in the potential temperature inversion height due to blocking and increased turbulence from the tall buildings. There is no indication of artifacts in these plots over the $z \in [480, 600]$m grid spacing transition region as well.

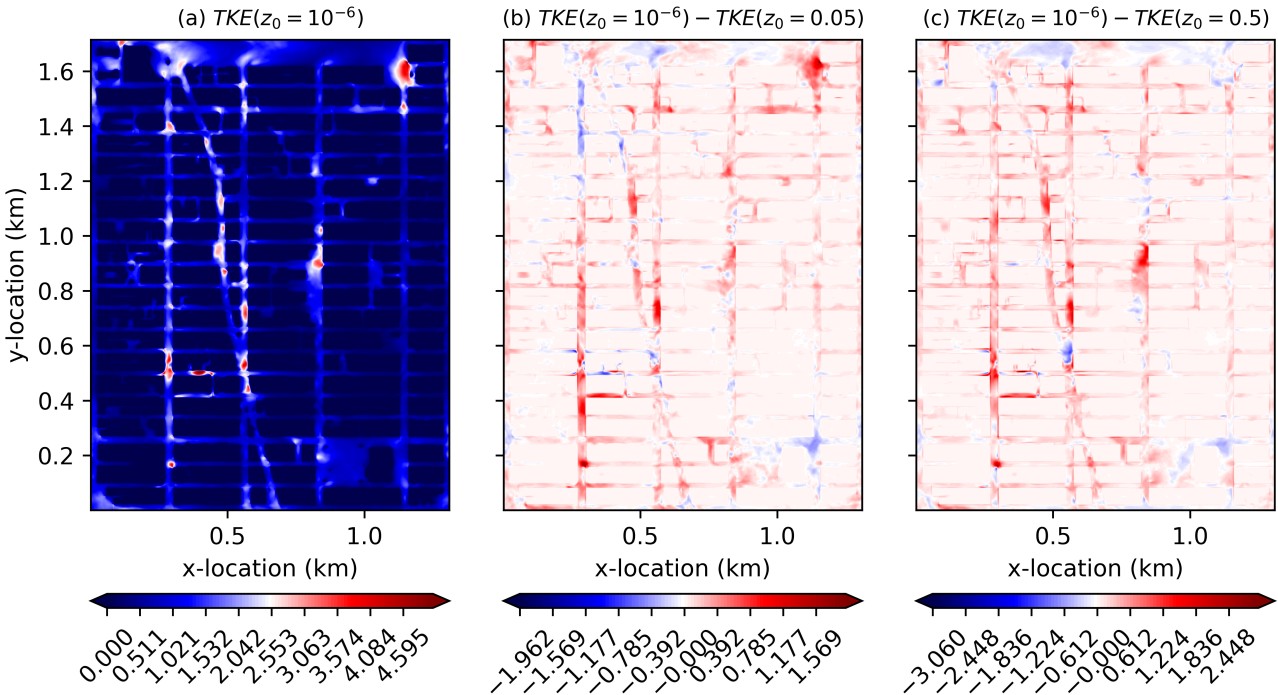

**Figure 18.** For the flow through Manhattan buildings test case, plots of 90-minute *resolved* TKE (perturbations are deviations from the time-average) and differences for different building roughness lengths at a height of $z = 1$m for flow through Manhattan buildings.

### 3.8 Performance

To get a sense of the performance of portUrb on GPUs compared to a similar model, the convective ABL simulation is performed with $2048 \times 2048 \times 122$ grid cells in the $x$, $y$, and $z$ directions, respectively with a grid spacing of 20m in all directions. The FastEddy model reports a performance of 7.3 time steps per wallclock second in Sauer and Muñoz-Esparza (2020) using 32 Nvidia V100 GPUs. Since access to this model of GPU is no longer available to the current authors, portUrb is run on Nvidia A100 GPUs and AMD MI250X GPUs using a number of GPUs that roughly equates to the same aggregate memory bandwidth. One could also make the choice to equate the aggregate compute throughput in terms of floating point operations per second (flop/s), but most GPU kernels in weather, climate, and LES models have relatively low arithmetic intensity. While the reconstruction kernels in portUrb are arithmetically intense, most other routines have the typical low arithmetic intensity of stencil-based fluids codes.

Each MI250X GPU is composed of two Graphics Compute Dies (GCDs), and therefore, the values for AMD MI250X GCDs will be used. The peak memory bandwidths of an Nvidia V100 GPU, an Nvidia A100 GPU, and an AMD MI250X GCD are 900 GB/s, 1555 GB/s, and 1638.4 GB/s, respectively. While this would suggest using 18 A100 GPUs and 18 AMD MI250X GPUs to give the aggregate memory bandwidth of 32 Nvidia V100 GPUs, an MPI decomposition of $3 \times 6$ tasks is less advantageous in terms of parallel data transfers than an MPI decomposition of $4 \times 4$ tasks. Therefore, 16 GPUs are used as a

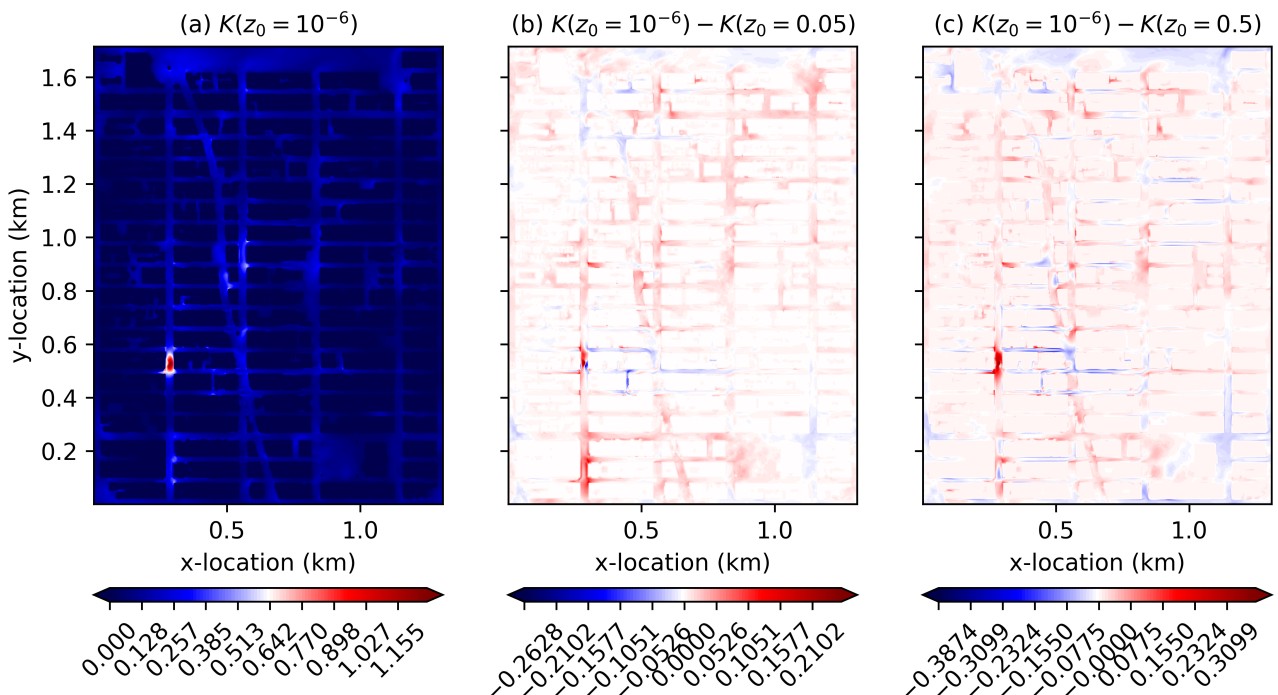

**Figure 19.** For the flow through Manhattan buildings test case, plots of 90-minute *unresolved, sub-grid-scale* TKE and differences for different building roughness lengths at a height of $z = 1$m for flow through Manhattan buildings.

comparison point in this study, even though the aggregate memory bandwidths are only 85-90% of the 32 V100 GPUs, putting this study's results at a slight memory bandwidth disadvantage. GPU-aware MPI is used in all simulations so that GPUs on separate MPI tasks are communicating directly from GPU memory rather than moving data to and from host memory. For sake of readability and simplicity, no effort is made in portUrb to use constant memory, texture caches, or user-controlled L1 cache.

The performance of portUrb in terms of time steps per wallclock second for different model configurations (order of accuracy and use of a Weighted Essentially Non-Oscillatory limiter) are given in Table 1. The fifth-order-accurate configuration without the WENO limiter is likely the most comparable to the FastEddy code and gives comparable performance on Nvidia A100 GPUs using similar aggregate memory bandwidth. On A100 GPUs, 9th-order accuracy is 36% more expensive than 5th-order accuracy without the WENO limiter and 53% more expensive with the WENO limiter. The AMD MI250X GCD is generally about 40% slower than the A100 using the same number of GCDs, even though the MI250X GCD has more memory bandwidth. This is likely due to the AMD HIP compiler showing higher register usage per kernel and having a small hardware managed L1 cache.

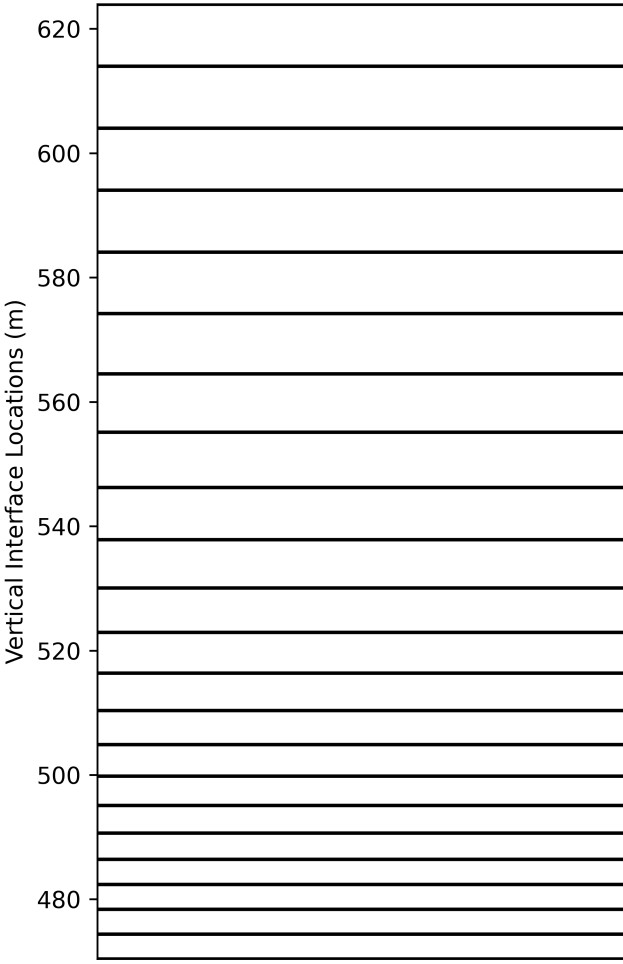

**Figure 20.** Vertical grid spacing over the transition region in the city flow forced by turbulent precursor test case.

|  | 9th, WENO | 9th, nolim | 5th, WENO | 5th, nolim |
|---|---|---|---|---|
| 16 A100 GPUs | 4.3 | 5.3 | 6.6 | 7.2 |
| 16 MI250X GCDs | 3.2 | 3.5 | 4.8 | 5.2 |

**Table 1.** Number of time steps simulated per wallclock second for the convective ABL *performance* test case using 2048×2048×122 grid cells. "9th" and "5th" indicate ninth-order accuracy and fifth-order accuracy in the reconstructions, respectively. "WENO" and "nolim" indicate use and absence of the WENO limiter in reconstruction. The baseline comparison point is that FastEddy completed 7.3 time steps per wallclock second for this problem size on 32 Nvidia V100 GPUs.

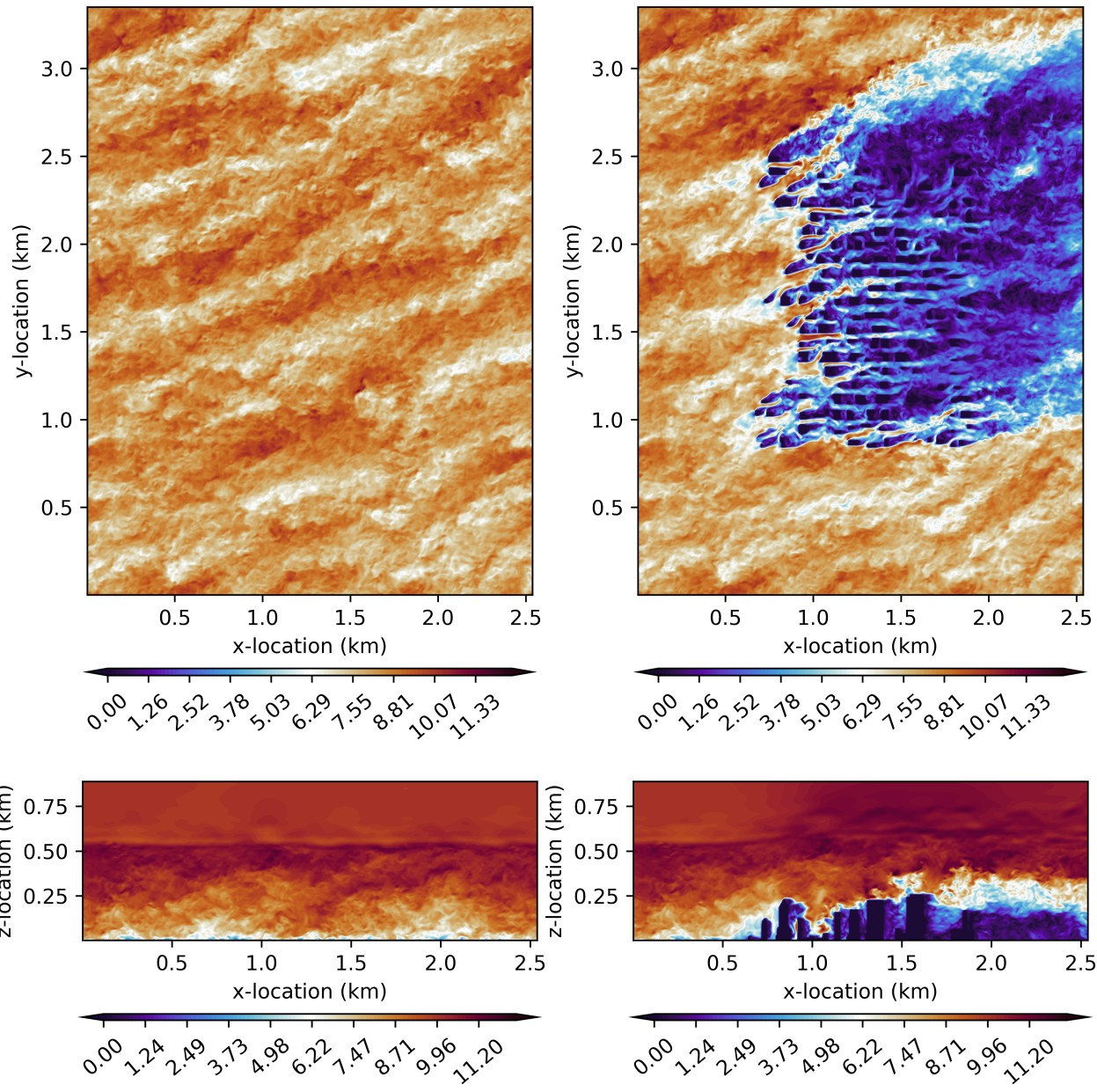

**Figure 21.** For the city flow forced by turbulent precursor test case, plot of instantaneous horizontal wind speed (m/s) at $t = 10$hr for the concurrent turbulent precursor simulation (left) and the forced simulation with the city included (right) at $z = 100$m (top) and at $y = 1,676$m (bottom).

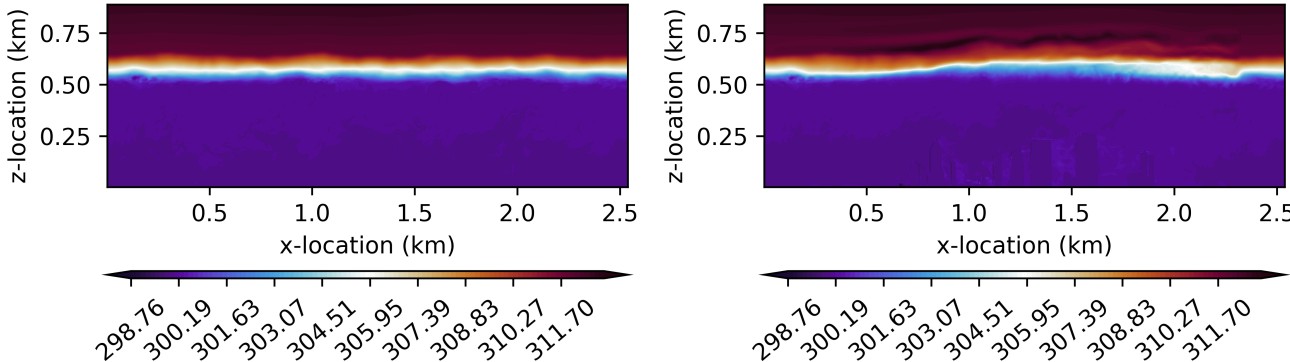

**Figure 22.** For the city flow forced by turbulent precursor test case, plot of instantaneous potential temperature (K) at $t = 10$hr for the concurrent turbulent precursor simulation (left) and the forced simulation with the city included (right) at $y = 1,676$m for the city forced by turbulent precursor simulation.

## 4 Concluding Discussion and Future Directions

The portUrb code presented here aims toward portability, performance, accuracy, simplicity, readability, robustness, extensibility, and ensemble capabilities for moist, atmospheric Large Eddy Simulation workflows with an emphasis on free-slip immersed boundaries with surface friction to account for urban building geometries. The use of portable C++ libraries enables portability and performance on CPUs as well as Nvidia, AMD, and Intel GPUs and future accelerated architectures. The use of eleventh-order Weighted Essentially Non-Oscillatory (WENO) interpolation enables accuracy and uses the compute capability of GPUs. The use of WENO limiting and upwind Riemann solvers improves robustness. The use of a simple and unambiguous coupler state definition eases extensibility. The inclusion of an ensemble capability that splits the MPI communicator and uses custom coupler settings for each ensemble member enables large ensemble groups within a single MPI program to more easily use Leadership Computing Facilities (LCFs) at capability scales. The model is currently cast in equal grid spacing Cartesian geometry with a variable vertical grid for simplicity and easy cell-finding with basic arithmetic. In all parts of the code, comments, clear looping structure, and clear operations seek to improve clarity and readability.

In section 3, a number of numerical experiments investigate the behavior of this mathematical formulation and implementation in several contexts. Neutral and convective atmospheric boundary layers demonstrate accuracy of the dynamical core, surface fluxes, and turbulent closure scheme by comparison against other LES models and against observations (in the convective case). A splitting supercell test case validates the moist configuration of the model against other models and microphysics formulations. Flow through cubes validates the frictioned immersed boundary approach against wind tunnel observations of mean values and turbulent fluctuations. Flow past a sphere is used to tune the partially immersed parameters to give accurate results at low resolution compared to high-resolution simulations in the mean flow statistics. Finally, flow through a section of buildings in Manhattan is used to investigate the influence of building roughness lengths in the frictioned immersed boundary

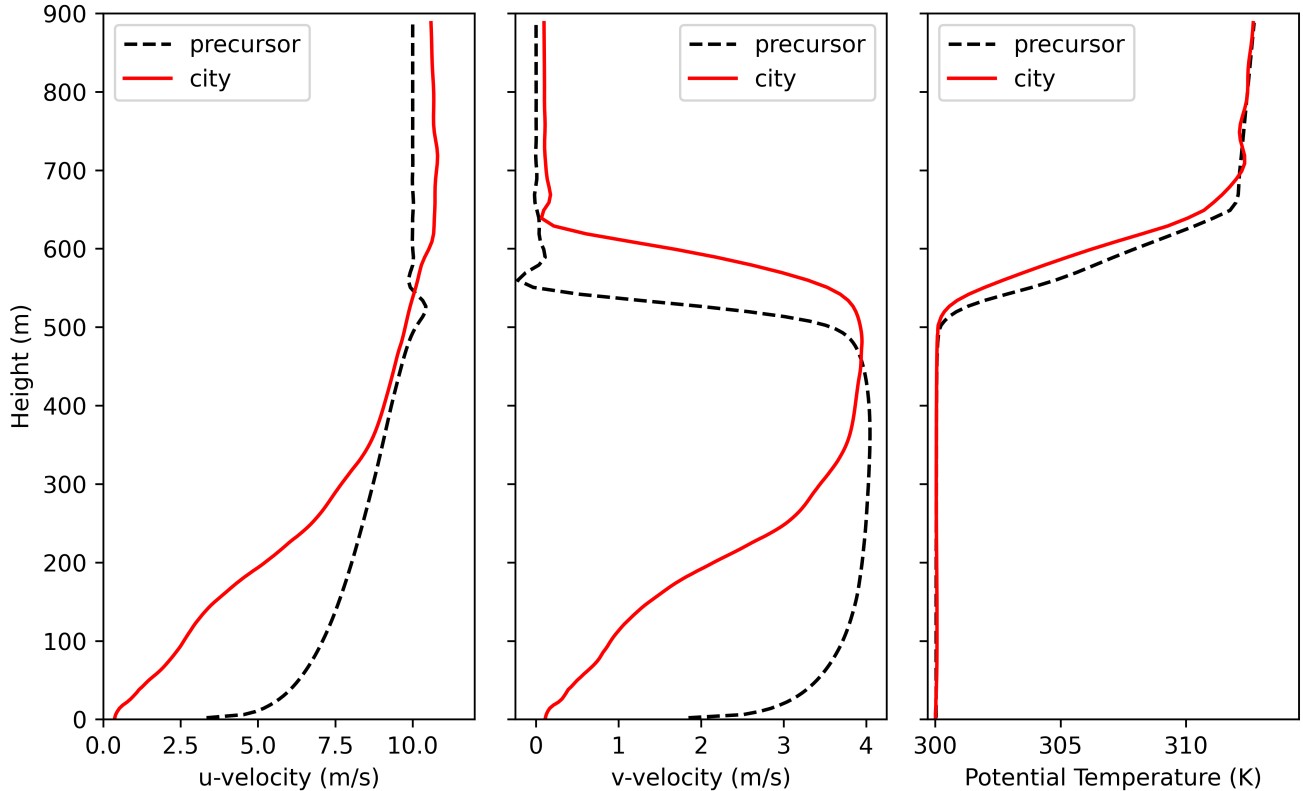

**Figure 23.** For the city flow forced by turbulent precursor test case, plot of the time-average over $t \in [8, 10]$hr of $u$-velocity in m/s (left), $v$-velocity in m/s (middle), and potential temperature in K (right) averaged over the city domain in the horizontal for precursor simulation and the city simulation forced by the precursor.

implementation. Those simulations demonstrate that even when directly ingesting complex building geometry from Open-StreetMap without smoothing or filtering, portUrb gives physically realizable results free of Runge oscillations and showing expected flow modifications associated with changing building surface friction.

     The most obvious room for improvement in portUrb is the handling of acoustic wave speeds in the dynamics, since most wind speeds are much lower than the speed of sound at the surface of roughly 350 m/s. The three most common possibilities

for this are sub-cycling of acoustics at a smaller time step ("split-explicit") (Skamarock and Klemp, 1992, 2008), anelastic and pseudo-incompressible approximations (Bannon, 1996; Durran, 1989; O'Neill and Klein, 2014), and semi-implicit treatment of acoustics (Weller et al., 2013; Gardner et al., 2018). Each of these options has multiple implementation choices, and therefore, this will be a significant effort to investigate the best fit. One challenge with infinite sound speed and time-implicit acoustic approximations is ensuring the immersed boundaries do not generate oscillations in the flow. This challenge is lower with

acoustic sub-cycling, and therefore, that option will likely be investigated first.

There are other options being considered for future research as well. The current portUrb code does not support terrain-following vertical coordinates. A metric transformation could be a helpful addition for domains where there is significant terrain. While immersed cells can be used to represent terrain quite easily, if the terrain varies significantly, this might lead to significant wasted computational effort. Having a smooth metric transformation for terrain-following vertical coordinates would not increase model complexity too appreciably and would reduce wasted computations. Immersed boundaries can be used to represent variations on the smooth overall terrain change over a domain represented by a vertical metric transformation. Metric transformation of the horizontal directions is likely not as helpful since the domains of simulation are intended to be relatively small with portUrb, and horizontal transformations would make cell-finding and horizontal derivative calculation more complex. While a vertical metric transformation will likely be worth that trade-off, a horizontal metric transformation might not.

Another capability planned for future implementation is two-way grid nesting. There are many possible approaches to improving resolution over part of a domain, including adaptive mesh refinement (Almgren et al., 2023) and stretched grids. These are all possibilities, but again, the complexity of the resulting model is of paramount consideration in a codebase mainly geared toward rapid prototyping of new model physics and generation of physics-based surrogate models. Nesting is a relatively low-overhead way to implement improved resolution, though it is certainly not without its challenges such as managing potentially different physics inside of nudged boundaries.

Intended applications of portUrb will make significant use of the ensemble and GPU acceleration capabilities to explore generation of training data for surrogate models of various atmospheric model processes like intra-cell microphysics, unresolved turbulence, super-resolution of coarse model results, and two-way communication of weather and urban models. The goal of future research with portUrb is to enable rapid prototyping of workflows for both offline and online training of Neural Network based surrogate models, efficient deployment of fine-grained surrogates, and understanding how surrogates behave when deployed in a model.

*Code and data availability.* The code is available at https://zenodo.org/records/15000787 (Norman, 2025a). Simulation results are available at https://zenodo.org/records/15232629 (Norman, 2025b).

*Author contributions.* Matthew R. Norman: Primary developer of portUrb and primary writer of this manuscript

Muralikrishnan Gopalakrishnan Meena: Co-developed the turbulence closure and surface friction scheme. Aided in creation of images. Aided in writing and editing the manuscript.

Kalyan Gottiparthi: Co-developed the turbulence closure and surface friction scheme. Aided in writing and editing the manuscript.

Nicholson Koukpaizan: Co-developed the turbulence closure and surface friction scheme. Co-developed the immersed boundary approach. Aided in writing and editing the manuscript.

Stephen Nichols: Co-developed the turbulence closure and surface friction scheme. Co-developed the immersed boundary approach. Aided in writing and editing the manuscript.

*Competing interests.* There are no competing interests for any of the authors.

*Acknowledgements.* This research used resources of the Oak Ridge Leadership Computing Facility at the Oak Ridge National Laboratory,
which is supported by the Office of Science of the U.S. Department of Energy under Contract No. DE-AC05-00OR22725.

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
