# Peer review of "PortUrb: A Performance Portable, High-Order, Moist Atmospheric Large Eddy Simulation Model with Variable-Friction Immersed Boundaries"

_EGUsphere, 2025_

## Referee Comment (RC2)

**Review of Geoscientific Model Development manuscript egusphere-2025-1135: "PortUrb: A Performance Portable, High-Order, Moist Atmospheric Large Eddy Simulation Model with Variable-Friction Immersed Boundaries" by Matthew Norman, Muralikrishnan Gopalakrishnan Meena, Kalyan Gottiparthi, Nicholson Koukpaizan, and Stephen Nichols.**

This article introduces and demonstrates the portUrb model, an LES code designed with the intention of providing portability, performance, accuracy, simplicity, readability, robustness, extensibility, and ensemble capabilities. The code is written in C++ and advertised to work both on CPUs and different GPU architectures (Nvidia, AMD, and Intel). A number of canonical ABL cases are presented to demonstrate and verify the adequacy of the model implementation, some of which include comparisons to other existing LES results from the literature. I want to particularly commend the authors for the great work in this new LES model development, and for the clarity with which the article is mostly written and organized. That is not an easy task when needing to convey such a large amount of information and details. I am supportive of seeing this manuscript published and give portUrb visibility for the community. However, there are a number of aspects the authors need to address before the paper can be published in GMD.

Major comments:

1) Figure 4 and associated discussion. Spectral energy distribution from Fig. 4 does not support the claim of 4-8dx effective resolution. It appears to be closer to ~8dx, even starting to deviate from the theoretical -5/3 for slightly larger scales than 8dx.
2) Section 3.3.2, Figs. 10 and 11. It would be desirable to include digitized results from Morrison and Milbrandt (2011) here for reference. It is otherwise very difficult to assess the comparison, and it makes really difficult to follow the discussion.
3) Section 3.5. I am not sure what this test case is adding from a code verification standpoint. I would suggest removing it unless results are compared to previous studies. In addition, the very limited discussion does not provide any insight that contributes to the purpose of the paper.
4) Section 3.6.
    a. Lines 495-496. This vertical domain extent does not allow for a proper interaction between the urban features and the ABL. This needs to be mentioned, even if for the purpose of simply running an urban case this may be okay.
    b. Lines 501-502. This is another questionable setup choice. I am not sure this test case serves as much of a purpose as it could be. At least using a deeper domain, including Coriolis effects, and considering some sort of at least idealized vertical profiles for BCs would be a better way to assess the objective (1) in the first paragraph of Sect. 3.6. In addition, I would strongly recommend the authors select a field campaign experiment itself (e.g., OKC Joint Urban 2003 is the most common), to in addition have the opportunity to validate their results against some observations.
5) Grid spacing. It seems like portUrban is restricted to using uniform grid spacing in all directions (this is not specifically mentioned but appears to be the case based on all the examples presented in the paper). While this is okay in the horizontal directions, not being able to have a vertically stretched grid is a serious drawback for ABL simulations. Otherwise, it will be very computationally expensive to cover the required vertical extent

while maintaining sufficiently fine grid spacings near the surface. I strongly encourage the authors to work on this enhancement as their priority if they want to see portUrb being of scientific value for atmospheric simulations and used by the community. In the current state, the code is significantly hindered in its applicability to relevant problems.

6) Another aspect that needs to be clearly acknowledged is the apparent limitation of portUrb to not include terrain representation capabilities. That would make it really challenging to apply it to realistic atmospheric problems, since there are always terrain variations present, and that in some cases are dominant ABL drivers.

7) In addition to points #4 and #5, portUrb seems to only allow for the use of laterally periodic boundary conditions. Nothing but highly idealized ABLs can be reproduced with such setup, significantly limiting the applicability to relevant real-world scenarios.

8) While this requires some non-negligible development, I strongly suggest the authors consider at least the incorporation of the capabilities mentioned in #4 and #5 before releasing the code (and ideally #6, although that could come with a future release). These two are basic capabilities without which I am not sure there is enough value to bring the model to the open community, since the use would be highly restricted.

9) Code 'performance' is mentioned several times as one of the key targets aimed with portUrb. However, this aspect is not assessed to any extent on the paper. I recommend the authors find some examples of code performance experiments in some of the atmospheric LES codes they mention in the introduction and report in more detail about timings and scaling across processors.

Other comments:

- Line 67. Later in the section (Eqs. 5 and 6) there is viscous terms, so this should be Navier-Stokes equations then.
- Eq. 1. Do the horizontal directions use total pressure and the vertical is perturbation pressure? If not the d/dx and d/dy terms need to have p' instead of p.
- Lines 157-158. Looks like this sentence requires some grammatical fixes.
- Lines 215-216. I wonder how sensitive are the results to the choice of the roughness at the immersed boundary surfaces, especially given the general lack to prescribe their values? It could be beneficial to investigate this sensitivity and include some brief summary in an appendix.
- Lines 271-272. I am not sure this adds anything since a uniform value of SGS TKE will not trigger any SGS TKE production by itself. Initial potential temperature perturbations will be more effective for that matter.

---

## Author Response (AR1)

**Revision 1: Response to Reviewers 1 and 2**

October 22, 2025

**1 Response to Reviewer 1**

We are grateful to Reviewer 1 for taking valuable time to read through the manuscript and request changes to improve the manuscript's quality, clarity, and usefulness to the readers of GMD. We are also thankful for the positive comments regarding the manuscript being well-written and streamlined. We respond to each of the comments below, including relevant changes to the revised manuscript that were made as a result of the comment.

1. *"In lines 55–58, the authors note that a series of numerical discretization studies led to their choice of discretization schemes. To enhance the manuscript's clarity and value to readers, it would be helpful if the authors briefly discussed the rationale behind selecting these particular schemes in the introduction. What specific advantages or limitations influenced their decision? Including this context—such as trade-offs in accuracy, stability, or computational cost—would strengthen the motivation and transparency of their methodological choices."*

   We agree that having this additional context will help the reader. We have added two new paragraphs to the introduction to provide this information, including why certain discretizations in previous publications like ADER-DT were not chosen for this study.

2. *"While the introduction emphasizes the portability of the model, it is unclear how this is demonstrated in the manuscript. Could the authors clarify what specific aspects of the code design or implementation contribute to its portability across architectures? Additionally, it would be helpful to include performance metrics—such as total runtime or computational resources used—for the Neutral Atmospheric and Dry Convective Boundary Layer test cases. This information would give readers a better understanding of the model's efficiency and scalability."*

   We agree that the two requests here are beneficial to the reader in understanding portability and performance in a more contextual and concrete manner. We have added text to the introduction in the revised manuscript to make it clear that it is the Kokkos portable C++ library that enables the same code to run efficiently on multiple GPU architectures (including Nvidia, AMD, and Intel) and CPU architectures.

   We have also added a new performance section to the results in the revised manuscript that details performance for the convective boundary layer test case. We report performance in terms of time steps computed per wallclock second because the other reviewer requested specifically a *comparative* performance benchmark that can be compared against other LES codes. This was the metric chosen in the most applicable performance benchmark we could find in literature. Given the hardware used in that paper is no longer widely available, we sought to match aggregate memory bandwidth (a common choice) in our comparison points for a true comparison between the two codes' performance and ran on two different vendors of GPU: Nvidia A100 and AMD MI250X.

3. *"It is not clear how geostrophic forcing enters Equation 1. Could the authors explicitly state how this term is incorporated into the governing equations? Additionally, in line 297, the statement that "Equation 18 is applied using the average column" is ambiguous. What does "average column" refer to?"*

We agree that this text is poorly worded and needs to be tied explicitly into the main PDE system. Therefore, we have reworked the text in this section to be more clear and have added a new equation to show how it relates to the continuous PDE system shown beforehand.

4. *"The manuscript contains several references to figures in other studies when comparing results. For instance, in line 310, the authors state that "these profiles closely match Figure 3 in Sauer and Muñoz-Esparza (2020)," which requires readers to consult external sources while reading. To improve readability and make the manuscript more self-contained, I recommend including the relevant figures from these referenced studies alongside the corresponding figures in this manuscript (e.g., overlaying Figure 3 from Sauer and Muñoz-Esparza (2020) with Figure 3 here). This approach would allow readers to directly assess the comparison without referring to external documents. A similar suggestion applies to Figure 2 and other instances where visual comparisons are made."*

We agree that the reader may find it frustrating to need to reference figures from other publications, and Reviewer 2 brought this up as well. While we cannot reproduce contour plots from other studies, we can extract line plot data and place them into plots we generate ourselves. Therefore, for Figures 2, 3, 7, 8, 10 and 11 all now have comparison points plotted from the other studies in question.

5. *"In line 306, the authors mentioned that "the smoothing of the second temperature discontinuity is lower in portUrb than seen in FastEddy". The author did not provide any evidence to support this statement."*

We agree that without comparison lines placed in the plots, there is no direct evidence for this statement in the publication, and this needs to be fixed. With the new Figure 2 in the revised manuscript, we believe there is evidence to show that the second potential temperature discontinuity at $z = 650$m is sharper in portUrb than in FastEddy. We also added the height at which the second temperature discontinuity occurs so that the text is more clear for the reader.

**2 Response to Reviewer 2**

We are grateful to Reviewer 2 for taking valuable time to provide a detailed and helpful review of the manuscript, and we feel the manuscript quality, clarity, and usefulness are greatly improved by this feedback. We are thankful for the positive comments regarding clarity, writing, and organization. We respond to each of the reviewer's comments below along with detailing resulting changes to the revised manuscript:

**2.1 Major Comments**

1. *"Figure 4 and associated discussion. Spectral energy distribution from Fig. 4 does not support the claim of 4-8dx effective resolution. It appears to be closer to ~8dx, even starting to deviate from the theoretical -5/3 for slightly larger scales than 8dx."*

We agree with this assessment and have changed the text in the revised manuscript to say that the effective resolution is about $8\Delta x$ and possibly a bit larger.

2. *"Section 3.3.2, Figs. 10 and 11. It would be desirable to include digitized results from Morrison and Milbrandt (2011) here for reference. It is otherwise very difficult to assess the comparison, and it makes really difficult to follow the discussion."*

We agree that this will greatly help the reader, and Reviewer 1 mentioned this regarding Figures 2 and 3 as well. We have now included baselines in Figures 2, 3, 7, 8, 10 and 11 so that the reader has them available in the manuscript itself.

3. *"Section 3.5. I am not sure what this test case is adding from a code verification standpoint. I would suggest removing it unless results are compared to previous studies. In addition, the very limited discussion does not provide any insight that contributes to the purpose of the paper."*

This is included in order to give the reader a detailed account of how we optimized the immersed boundary forcing for partially immersed cells in a data-driven manner. While it may not be validation per se, we hope it helps the reader understand concretely how this optimization was performed. We understand the request to remove it. Still, we hope to keep it in to provide what we believe are useful design details for the reader.

4. *a. Lines 495-496. This vertical domain extent does not allow for a proper interaction between the urban features and the ABL. This needs to be mentioned, even if for the purpose of simply running an urban case this may be okay.*
   *b. Lines 501-502. This is another questionable setup choice. I am not sure this test case serves as much of a purpose as it could be. At least using a deeper domain, including Coriolis effects, and considering some sort of at least idealized vertical profiles for BCs would be a better way to assess the objective (1) in the first paragraph of Sect. 3.6. In addition, I would strongly recommend the authors select a field campaign experiment itself (e.g., OKC Joint Urban 2003 is the most common), to in addition have the opportunity to validate their results against some observations.*

   We agree that including a higher vertical top, a realistic forcing, and Coriolis effects is beneficial for this kind of test case. In order to address these requests, we have added a new test case (Section 3.7 in the revised manuscript) that forces the city simulation with a concurrent turbulent precursor simulation based on the same initial conditions and forcings used in the neutral ABL test case (an inversion at 500m and geostrophic forcing for an Ekman spiral). For boundaries for which normal velocity amounts to inflow, the precursor data is copied into the ghost cells for each Runge Kutta stage (saved from the precursor simulation); and for outflow boundaries, open boundaries are used. We also use 4m grid spacing in the horizontal and a stretched vertical grid that transitions from 4m grid spacing to 10m grid spacing over the transition domain $z \in [480, 600]$m so that the city buildings use fine grid spacing, and the ABL top uses coarser grid spacing. This simulation and the results demonstrate how mechanically forced turbulence from the buildings act to mix the inversion layer and entrain the warmer potential temperature that forms the inversion layer. We agree that a field campaign like the OKC Joint Urban 2003 experiment would be ideal. However, the army.mil link to the actual data has been inaccessible for us from all machines and networks we have tried over the last several years. Therefore, while the data was likely open to the public at one point, it appears not to be available to the public now.

5. *"Grid spacing. It seems like portUrban is restricted to using uniform grid spacing in all directions (this is not specifically mentioned but appears to be the case based on all the examples presented in the paper). While this is okay in the horizontal directions, not being able to have a vertically stretched grid is a serious drawback for ABL simulations. Otherwise, it will be very computationally expensive to cover the required vertical extent while maintaining sufficiently fine grid spacings near the surface. I strongly encourage the authors to work on this enhancement as their priority if they want to see portUrb being of scientific value for atmospheric simulations and used by the community. In the current state, the code is significantly hindered in its applicability to relevant problems."*

   The reviewer is correct that the version of portUrb in the original manuscript used uniform grid spacing in all directions, and we agree with the reviewer that this is likely too limiting for practical use by LES and ABL modelers. We have implemented a variable vertical grid using a metric transformation to and from a uniformly spaced grid, which is described in the revised manuscript. Also, we exercise this capability in the new test case added in Section 3.7 by transitioning the vertical grid spacing from 4m to 10m over a domain of 120m. We have also tested the new variable vertical grid in unit tests, comparing the solution with and without the use of this capability, and it passes all integration tests. In the simulations shown in this paper, there are no artifacts in the transition region.

6. *"Another aspect that needs to be clearly acknowledged is the apparent limitation of portUrb to not include terrain representation capabilities. That would make it really challenging to apply it to realistic atmospheric problems, since there are always terrain variations present, and that in some cases are*

*dominant ABL drivers.*"

We have text in the last section of the submitted manuscript that says, "A metric transformation could be a helpful addition for domains where there is significant terrain. While immersed cells can be used to represent terrain quite easily, if the terrain varies significantly, this might lead to significant wasted computational effort. Having a smooth metric transformation for terrain- following vertical coordinates would not increase model complexity too appreciably and would reduce wasted computations. Immersed boundaries can be used to represent variations on the smooth overall terrain change over a domain represented by a vertical metric transformation."

In response to the reviewer's concern, we have added to the revised manuscript the sentence, "The current portUrb code does not support terrain-following vertical coordinates." preceding the afore-mentioned text. We hope the existing text along with this addition will make it clear to the reader that this is a current limitation of portUrb.

7. *"In addition to points #4 and #5, portUrb seems to only allow for the use of laterally periodic boundary conditions. Nothing but highly idealized ABLs can be reproduced with such setup, significantly limiting the applicability to relevant real-world scenarios."*

We agree with the reviewer that we did not properly specify in the submitted manuscript what boundary conditions portUrb is capable of supporting. We have added to the revised manuscript a discussion that portUrb supports forced (precursor), open, solid wall, and periodic boundary conditions. We demonstrate the forced and open boundary conditions for the reader in the new section 3.7 experiment that forces flow past a city with a concurrent turbulent precursor based on geostrophically forced neutrally stratified boundary layer flow with an inversion beginning at 500m.

8. *"While this requires some non-negligible development, I strongly suggest the authors consider at least the incorporation of the capabilities mentioned in #4 and #5 before releasing the code (and ideally #6, although that could come with a future release). These two are basic capabilities without which I am not sure there is enough value to bring the model to the open community, since the use would be highly restricted."*

We believe that the reviewer's request is quite reasonable, though it did indeed require some significant development. The boundary conditions were implemented already (we simply neglected mentioning it in the submitted manuscript), but we have now implemented variable vertical grid spacing and integrated its use into internal integration tests to ensure they remain correct. Terrain following coordinates are slated for future development, but terrain can currently be supported by use of immersed and partially immersed boundaries, since immersed boundaries have a roughness length specified at each immersed cell.

9. *"Code 'performance' is mentioned several times as one of the key targets aimed with portUrb. However, this aspect is not assessed to any extent on the paper. I recommend the authors find some examples of code performance experiments in some of the atmospheric LES codes they mention in the introduction and report in more detail about timings and scaling across processors."*

We agree with the reviewer (and reviewer 1) that this is helpful information for the reader. Since a comparative assessment was requested, we sought the closest comparisons we could find, and this ended up being from the FastEddy 2020 paper, Figure 18, which uses the convective ABL test case along with a certain number of horizontal grid cells at a fixed grid spacing of 20m on 32 Nvidia V100 GPUs. We have detailed this test case in a new results section and provided a comparison against the largest test case in the FastEddy paper against two different types of GPUs (Nvidia A100 and AMD MI250X), roughly equating the total aggregate peak memory bandwidth between our experiments and FastEddy's experiment. We also tested four configurations of portUrb for this test case: 9th-order with WENO, 9th-order without WENO, 5th-order with WENO, and 5th-order without WENO so that the reader can have some sense of what the slowdown associated with high-order and WENO limiting are

separately and potentially make different choices than this study has based on their throughput needs.

**2.2 Other Comments**

1. *"Line 67. Later in the section (Eqs. 5 and 6) there is viscous terms, so this should be Navier- Stokes equations then."*

   We agree, and we have made this change in the revised manuscript

2. *"Eq. 1. Do the horizontal directions use total pressure and the vertical is perturbation pressure? If not the d/dx and d/dy terms need to have p' instead of p."*

   We do indeed use perturbation pressure in the horizontal as well, and we have made the requested change in the revised manuscript. It has little numerical effect, as WENO weights ignore the mean value of a stencil, but we agree with the reviewer that it's important to be precise in the manuscript and its agreement with the code.

3. *"Lines 157-158. Looks like this sentence requires some grammatical fixes."*

   We agree the wording was confusing, and we have reworded this to be more clear for the reader.

4. *"Lines 215-216. I wonder how sensitive are the results to the choice of the roughness at the immersed boundary surfaces, especially given the general lack to prescribe their values? It could be beneficial to investigate this sensitivity and include some brief summary in an appendix."*

   We agree that this is an interesting aspect to investigate. It is our hope that the investigation of different building friction length choices in the experiments in Section 3.6: Flow Through Manhattan Buildings will provide the reader with a sense of how the results change when the immersed boundary roughness changes.

5. *"Lines 271-272. I am not sure this adds anything since a uniform value of SGS TKE will not trigger any SGS TKE production by itself. Initial potential temperature perturbations will be more effective for that matter."*

   We agree that this is a subtle issue. Notice in Equation (8) of the original and revised manuscripts that all SGS TKE sources and sinks depends on eddy viscosity, $K_m$, or directly on SGS TKE, $K$, particularly the shear production. By equation (6), the eddy viscosity is proportional to the square root of SGS TKE. Therefore, if SGS TKE is zero initially everywhere, there can be no shear production or buoyancy source production in convective regimes. So initial SGS TKE must be seeded. We agree that initial SGS TKE does not trigger production by itself, but a non-zero value is required for production to occur by other means.